# DE NOVO ANTIBIOTIC-LIKE MOLECULE DESIGN VIA DESCRIPTOR-GUIDED PATCH-BASED GANS

## ABSTRACT

The rise in antimicrobial resistance necessitates rapid discovery of novel antibiotics, but the majority of generative pipelines are unable to produce clinically viable candidates. The structural and topological complexity of actual antibiotics is not captured by current models, which lack pharmacological depth and are frequently trained on datasets such as QM9 and ZINC. We present a modular, descriptor-guided framework for antibiotic-like molecule design that combines a property-aligned $\beta$-VAE for interpretable encoding, a descriptor-to-latent conditioner for controllable sampling, and a patch-based graph generator for fragment-wise synthesis. Trained on a curated subset of ChEMBL containing clinically validated antibiotics, our framework supports end-to-end generation from RDKit descriptors to final molecules, with adversarial refinement for topological realism. Beyond favorable ADMET profiles, our method establishes *joint satisfaction reporting* of drug-likeness thresholds as a benchmark standard, resists mode collapse under 50k-sample stress tests, and surfaces ligands that outperform ciprofloxacin and co-crystal references in docking assays. These results highlight chemically meaningful, pharmacologically informed generation − overcoming limitations of black-box pipelines and general-purpose datasets.

## 1 INTRODUCTION

Antimicrobial resistance continues to pose threat to global health by decreasing the effectiveness of numerous antibiotics Ferrara et al. (2024). The development of new antibiotics has slowed considerably in spite of this urgency, in part due to the high failure rate and high cost of traditional small-molecule drug development. Only 1 in 15 candidates from current classes − and 1 in 30 from new classes − ever reach patients, according to the World Health Organisation, which states that it takes 10 to 15 years to bring an antibiotic from preclinical stages to market WHO (2022). As a result, the application of AI to accelerate development of pharmacologically viable and structurally unique antibiotic candidates is receiving increased attention Swanson et al. (2024).

Generative Adversarial Networks (GANs) are established tools for exploring large chemical spaces and generating new molecules with desired properties for de novo drug design Tong et al. (2021). Most generative models in use, however, are evaluated using datasets such as QM9 Ramakrishnan et al. (2014) and ZINC Tingle et al. (2023), which, although computationally efficient, do not provide the pharmacological depth required for practical antibiotic discovery Cauchy et al. (2023). ZINC concentrates on docking-ready compounds without experimental target or mechanism data, whereas QM9 consists of small, synthetically enumerated molecules without bioactivity annotations Glavatskikh et al. (2019). In contrast, ChEMBL offers a manually curated repository of drug-like molecules with extensive annotations including $IC_{50}$, Ki, mechanisms of action, and therapeutic targets Zdrazil et al. (2023). We focus on the generation of antibiotic-like molecules using a curated subset of ChEMBL consisting of clinically validated antibiotics.

To this end, we propose a property-aligned generative framework that conditions latent molecule representations on RDKit descriptors Landrum (2013), including quantitative estimate of drug-likeness (QED), the octanol-water partition coefficient (logP), and the synthetic accessibility score (SA). These descriptors are mapped to latent vectors and used to generate fixed-size graph patches − interpretable molecular fragments − that are subsequently assembled into full molecules. Unlike prior black-box GAN pipelines De Cao & Kipf (2018); Guimaraes et al. (2017); Wei et al. (2023),

which optimize or report one property at a time, our framework establishes *joint satisfaction of pharmacologically meaningful thresholds* as a new benchmark. Specifically, we report the fraction of molecules simultaneously meeting QED $> 0.6$, SA $< 5$, and logP $\in [-0.5, 5.0] -$ a population-level metric absent from existing baselines. Our descriptor-conditioned patch generator not only provides interpretable fragment-level control but also mitigates the mode collapse common in atom-level GANs Tang et al. (2024). By this, we introduce *multi-property alignment* as an evaluation metric, which helps yield hundreds of topologically diverse, jointly optimized antibiotic-like molecules with favorable ADMET profiles Daoud et al. (2021). The main contributions of this work are:

- A curated dataset of 4,607 antibiotic-like molecules from ChEMBL, annotated with drug-likeness properties and RDKit descriptors.

- A modular framework combining a $\beta$-VAE, descriptor-to-latent conditioner, and patch-based generator for interpretable, property-guided molecule design.

- Establishing *joint satisfaction reporting* (QED $> 0.6$, SA $< 5$, logP $\in [-0.5, 5.0]$) as a practical benchmark for multi-property drug design, exposing gaps in prior baselines.

- A 50k-sample stress test with 72% unique scaffolds was used to demonstrate robustness and confirm resistance to mode collapse.

- Establishing biological plausibility by using QSAR screening to find drug-like actives and docking against antibacterial targets (DHFR, DNA gyrase), where several produced ligands perform better than co-crystal references and ciprofloxacin.

## 2 BACKGROUND & RELATED WORK

**Fragment- & Graph-Based Models:** SMILES-based sequence models, such as RNNs and Transformers, suffered from syntactic invalidity and lacked structural interpretability Andronov et al. (2025). While graph-based models, such as MolGAN De Cao & Kipf (2018) and GraphAF Shi et al. (2020), demonstrated improved chemical validity via direct molecular graph generation, most of them suffered mode collapse and failed to provide fine-grained control over structural components. Recent diffusion-based models such as EDM Hoogeboom et al. (2022) showed strong performance in generating 3D molecular structures with high stability. While powerful, these models were primarily unconditioned and focused on geometric fidelity rather than pharmacological control. They lacked mechanisms for descriptor-level property steering and did not operate in a fragment- or scaffold-based regime. To introduce modularity, hierarchical models such as Junction Tree (JT)- VAE Jin et al. (2020) and DeepMGM Bian & Xie (2022) decomposed molecules into scaffold–fragment pairs, enabling improved scaffold retention. These models, however, often relied on fixed fragment vocabularies. ScaffoldGVAE Hu et al. (2023) introduced multi-view graph encoding to enhance scaffold-level diversity and pharmacophoric consistency.

**Domain-Specific Molecular Design:** Most existing generative models applied rewards post hoc $-$ Mol-CycleGAN Maziarka et al. (2020) and InstGAN Tang et al. (2024) optimized generated molecules towards target properties using reinforcement learning or adversarial training. However, they lacked explicit alignment between input descriptors and molecular representations. This shortcoming has made in-generation property control a key challenge in molecule generation. While most existing models were benchmarked on general-purpose datasets such as ZINC and QM9 Glavatskikh et al. (2019), recent efforts explored target-specific molecule generation. For example, Deep-MGM Bian & Xie (2022) focused on CB2-targeted libraries, and RuSH Rossen et al. (2024) introduced a reinforcement learning strategy for scaffold hopping based on pharmacophoric similarity.

Despite recent advances, key limitations persist in the molecular generation literature. First, most models prioritize structural validity and property optimization but lack mechanisms for interpretable, descriptor-guided control during generation Du et al. (2024). Second, fragment-based methods often rely on discrete vocabularies or scaffold libraries, limiting generative flexibility Voloboev (2024). Third, while some models incorporate property rewards, few align molecular latent spaces explicitly with interpretable chemical descriptors Haddad et al. (2025). Finally, not many works focus on antibiotic discovery, where new chemical diversity is urgently needed Schuh et al. (2025); Chen et al. (2023). This motivates the development of a modular, property-aligned framework that supports controllable, fragment-level generation tailored to a clinically relevant therapeutic space.

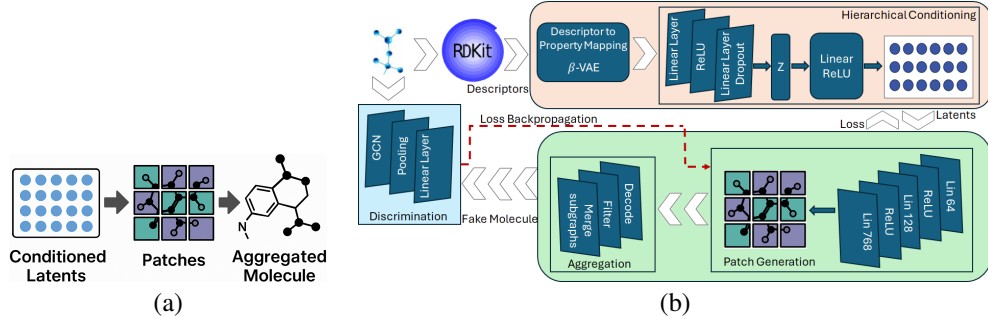

Figure 1: (a) Proposed molecule construction, (b) End-to-end framework overview

# 3 PROPOSED FRAMEWORK

## 3.1 PRELIMINARIES

**Inspiration and Adaptation:** Our design is inspired by modular, property-aware molecule generation concept. Fragment-based models such as JT-VAE Jin et al. (2020) and DeepMGM Bian & Xie (2022) motivate our patch-level synthesis approach; we adopt a continuous latent representation instead of fixed vocabularies though (Fig. 1(a)). Inspired by Mol-CycleGAN Maziarka et al. (2020) and InstGAN Tang et al. (2024), we employ $\beta$-VAE Higgins et al. (2017) to align learned latents with RDKit descriptors for controllable generation. Finally, ScaffoldGVAE Hu et al. (2023) inspires scaffold diversity and coherent graph reconstruction in our framework.

**Comparison with Related Works:** Table 1 summarizes key attributes for various molecule generation baselines. While prior methods offer strong validity or partial property control, few support fragment-based decoding, latent interpretability, or clinically relevant descriptor conditioning. In contrast, our framework integrates these capabilities into a unified pipeline aligned with ADMET evaluation and scaffold-aware synthesis.

Table 1: Comparison of molecular generation methods across key modeling properties

| Model | Dataset | Validity | Property Control | Latent Interpretability | Patch-Based Generation |
|---|---|---|---|---|---|
| MolGAN[*] | ZINC | ∼98% | ✗ | ✗ | ✗ |
| GraphAF[†] | QM9 | 100% | ✓(flow) | ✗ | ✗ |
| DeepMGM[‡] | ZINC | 100% | ✓ | ✓(vocab) | ✗ |
| EDM[§] | QM9 | 100% | ✗ | ✗ | ✗ |
| RuSH[¶] | ZINC | ∼99% | ✓(target-based) | ✗ | ✗ |
| HierVAE[ζ] | Polymers | 100% | ✗ | ✗ | ✓(Motif) |
| Proposed | ChEMBL | 100% | ✓(desc-cond) | ✓($\beta$-VAE) | ✓ |

**Legend:** [*]De Cao & Kipf (2018), [†]Shi et al. (2020), [‡]Bian & Xie (2022), [§]Hoogeboom et al. (2022), [¶]Rossen et al. (2024), [ζ]Jin et al. (2020)

**Dataset Construction and Feature Extraction:** We curate a representative dataset of antibiotic molecules from ChEMBL Zdrazil et al. (2023) by filtering drug indications for the keyword "antibiotic". Molecules with invalid SMILES or fewer than 10 heavy atoms are removed, yielding a high-quality set of 4,613 compounds. For each molecule, we compute three core properties: QED, logP and SA. These properties are calculated using RDKit and a publicly available SA scoring module. Each molecule $m$ is represented by a 3-dimensional feature vector:

$$\mathbf{x}_m = [\text{QED}(m),\ \text{logP}(m),\ \text{SA}(m)]$$

We then enrich this filtered set with 200+ RDKit descriptors Landrum (2013) including physicochemical and topological features. Let $\mathbf{d}_m \in \mathbb{R}^D$ denote the RDKit descriptor vector and $\mathbf{p}_m \in \mathbb{R}^3$ the property vector, then the final descriptor matrix is:

$$\mathbf{x}_m = [\mathbf{d}_m \parallel \mathbf{p}_m] \in \mathbb{R}^{D+3}$$

Although trained on antibiotics, our framework generalizes to any therapeutic class for which meaningful descriptors are available.

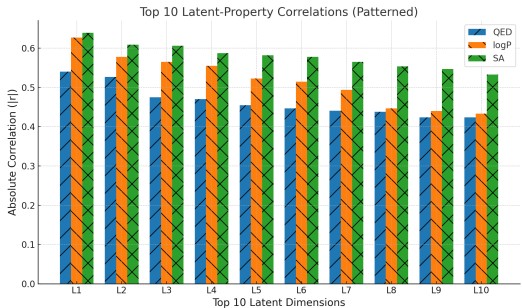

Figure 2: Latent dim. & mol. props. correlation

### 3.2 PIPELINE COMPONENTS

Fig. 1(b) presents an end-to-end framework showing how RDKit descriptors are mapped to latent property vectors via a $\beta$-VAE and Conditioner. These guide patch generation and aggregation into molecules, later evaluated by a graph discriminator. Adversarial feedback refines patch synthesis.

**Variational Autoencoder for Latent–Property Modeling:** To model molecular representations with interpretable, property-aligned latents, we train a $\beta$-VAE Higgins et al. (2017) with a supervised head for QED, logP, and SA prediction. The encoder is a 2-layer GIN Xu et al. (2019), which projects input graphs to latent parameters $\mu, \log \sigma^2$, for latent vector sampling:

$$z \sim \mathcal{N}(\mu, \mathrm{diag}(\sigma^2)), \quad z = \mu + \sigma \odot \epsilon$$

The decoder (MLP) reconstructs molecular patches, while an auxiliary head regresses property values. Total loss is:

$$\mathcal{L} = \beta \cdot \mathcal{L}_{\mathrm{KL}} + \lambda \cdot \mathcal{L}_{\mathrm{prop}}, \quad \mathcal{L}_{\mathrm{prop}} = \sum \mathrm{MSE}_{\mathrm{QED, logP, SA}}$$

with $\beta = 0.1$, $\lambda = 5.0$. Pearson correlations between latent dimensions and molecular properties (Fig. 2) reveal interpretable alignment. We extract the top-$k$ most correlated latents per property to support targeted conditioning and traversal. To ensure alignment between VAE latents and descriptors, we intersect molecules present in both representations. Invalid features or those with >99% constant values are removed, yielding a clean matrix $X \in \mathbb{R}^{n \times d}$ − used for downstream analyses:

$$X = \begin{bmatrix} x_1^\top & \cdots & x_n^\top \end{bmatrix}^\top, \quad n = 4607, \ d = 188$$

**Learning Descriptor-to-Latent Mappings via Conditioning:** To enable property-driven molecule generation, we train a *Conditioner* − a 3-layer MLP mapping RDKit descriptors $\mathbf{x} \in \mathbb{R}^d$ to a 9-dimensional latent vector $\mathbf{z}_{\mathrm{top}} \in \mathbb{R}^9$ aligned with molecular properties. These 9 latent dimensions are selected based on the highest Pearson correlations with QED, logP, and SA.

$$f_{\mathrm{cond}} : \mathbf{x} \mapsto \mathbf{z}_{\mathrm{top}}, \quad \mathbf{z}_{\mathrm{top}} = W_3 \cdot \mathrm{ReLU}(W_2 \cdot \mathrm{ReLU}(W_1 \cdot \mathbf{x}))$$

We train the conditioner using MSE loss with Adam (lr $= 10^{-3}$) for 50 epochs. The final model achieves a mean cosine similarity of 0.8773 between predicted and true latent targets, validating its ability to bridge descriptors and interpretable property-aligned embeddings. This enables molecule generation directly from descriptor space, without requiring the original graph. Although standard, the training architectures for $\beta$-VAE and conditioner are presented in Appendix Fig. 8(a) and (b).

**Patch Generator via Latent Supervision:** We train a patch generator $\mathcal{G}$ to synthesize node-feature matrices from descriptor-derived latent vectors. The generator is a 3-layer MLP that receives a 9-dimensional vector $\mathbf{z}_{\mathrm{top}}$ from Conditioner, and outputs a latent graph patch $\hat{\mathbf{P}} \in \mathbb{R}^{48 \times 16}$. To supervise training, we pass $\mathbf{z}_{\mathrm{top}}$ through a fixed decoder from the pretrained VAE to obtain a reference patch. The generator minimizes the MSE loss:

$$\mathcal{L}_{\mathrm{gen}} = \left\| \mathcal{G}(\mathbf{z}_{\mathrm{top}}) - \hat{\mathbf{P}} \right\|_2^2$$

We train for 25 epochs with Adam (lr $= 10^{-3}$), saving the best model based on validation loss. This supervised generator bridges tabular descriptors and molecular substructures, enabling controllable synthesis of latent graph embeddings.

**Patch Aggregation and Molecule Reconstruction:** To convert latent patches into molecules, we retain a dual-path decoding strategy. Each patch is a 2D tensor of atom embeddings; active rows represent atom candidates. Two decoding methods are applied and the candidate with the higher composite score is selected. The first approach builds a connectivity-first scaffold: nearby embeddings form single bonds under atom caps and degree limits, with six-membered cycles detected and protected. A fraction of short bonds are upgraded to double, and C/N six-rings undergo aromatization in one or two passes. Sanitization is performed without kekulization, and only the largest fragment is kept, with a single-bond fallback if the molecule collapses. The second approach interprets the patch as coordinate distances, forming bonds heuristically with RDKit before applying the same upgrade, aromatization, and fragment steps. If both decoded graphs are valid, we choose the one with the higher score: $\text{Score} = \text{QED} - 0.05 \cdot \text{SA} + 0.01 \cdot \text{logP}$

Only the highest-scoring valid molecule is retained, with its SMILES string and scores logged.

**Graph-Based Discriminator for Realism Detection:** To distinguish ChEMBL molecules from generated ones, we train a graph-based binary discriminator using a two-layer graph convolutional network (GCN) followed by global mean pooling and fully connected layers in a standalone warm-up phase with equal branches of the two types of graphs. Atom-level features and edge indices are derived from SMILES strings using RDKit and PyTorch Geometric. Given input $\mathbf{x}$ and edge set $\mathcal{E}$:

$$\mathbf{h}_1 = \text{ReLU}(\text{GCNConv}_1(\mathbf{x}, \mathcal{E})), \quad \mathbf{h}_2 = \text{ReLU}(\text{GCNConv}_2(\mathbf{h}_1, \mathcal{E}))$$

The pooled embedding is passed through two linear layers to produce a logit, optimized with binary cross-entropy:

$$\mathcal{L}_{\text{disc}} = \text{BCEWithLogitsLoss}(\hat{y}, y)$$

The pretrained discriminator serves both as a standalone molecule filter and as a feedback module in adversarial training. Intermediate graph embeddings are also used for diversity analysis.

**Adversarial Fine-Tuning through GAN Training:** We employ GAN-based training to improve patch generation, where the generator minimizes binary cross-entropy loss to learn to produce molecules that are indistinguishable from the real samples − as determined by the discriminator:

$$\mathcal{L}_{\text{gen}} = \text{BCE}(D(G(z)), 1)$$

where latent patches $z$ are aggregated into molecules and scored by the discriminator during each iteration, which alternates between discriminator- and generator-updates. This loop maintains property alignment while enhancing topological realism. Algorithm 1 in Appendix A.3 describes the framework's operation in a four-phase VAE-GAN pipeline.

## 4 RESULTS & DISCUSSIONS

All experiments are conducted using a fixed random seed for reproducibility. We use a curated subset of 4,607 ChEMBL antibiotic SMILES as input space, generating ∼9,000 candidate molecules per experiment. Chemical validity, uniqueness, and novelty are assessed with RDKit, alongside physicochemical properties (QED, logP, SA). Beyond these core metrics, we evaluate ADMET profiles (SwissADME/pkCSM heuristics), perform large-scale robustness tests, and conduct biological validation via docking (AutoDock Vina) and QSAR screening.

**Diversity and Latent Structure Diagnostics:** To assess whether the generator's output diversity reflects meaningful conditioning (not random dispersion), we compare three conditions: (1) untrained generator initialized with random weights, (2) trained generator receiving random latent vectors (with the conditioner removed), and (3) the full pipeline with descriptor-conditioned latent input. For each case, we visualize the t-SNE embeddings and cosine similarity heatmaps of the generated patches, see Fig. 3 and Appendix A.7. The untrained generator produces patches that are highly diverse (cosine similarity spanning 0.3–0.9), but the diversity is unstructured, as shown in the t-SNE scatter. This confirms that architectural priors alone are insufficient for semantically meaningful generation. In the ablation condition (trained generator with random latent vectors), patches remain diverse, but no consistent gradient is observed with respect to QED, logP, or SA. Only in the fully conditioned case do the patch embeddings exhibit both diversity and alignment with target properties, supporting the necessity of the conditioner for controlled, property-aware

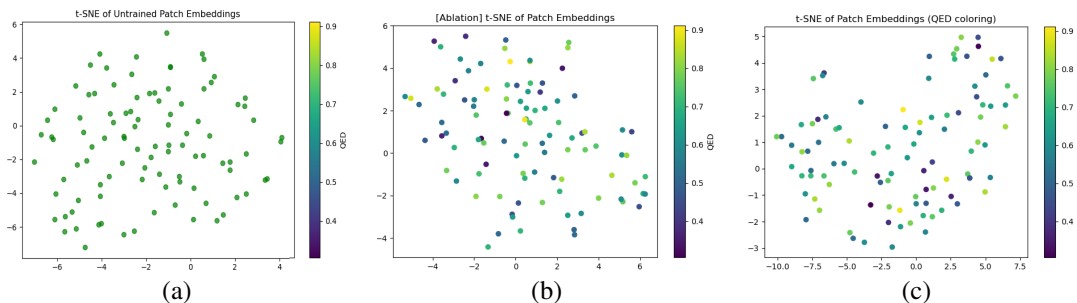

Figure 3: t-SNE embeddings of generated patch representations colored by QED score. (a) untrained generator, (b) trained generator, and (c) full pipeline with descriptor-conditioned latent input.

synthesis. Topological diversity and complexity metrics (Appendix Fig. 12) further confirm the generator's ability to produce pharmacologically realistic compounds. Note that we do not retrain the full GAN pipeline without the conditioner, since the ablation of property alignment is most directly attributable to the latent-level input. Adding GAN dynamics will introduce orthogonal sources of instability and obscure the conditioner's contribution.

**GAN Checkpoint:** To identify the optimal checkpoint, we analyze property scores for molecules generated per epoch, and develop a custom reward function to rank epochs by molecular quality:

$$\text{Reward} = 2 \times \text{QED} - 0.6 \times \left( \frac{\max(\text{SA} - 3, \ 0)}{4} \right) - 0.4 \times \left( \frac{\max(|\text{logP} - 2.5| - 1, \ 0)}{2} \right)$$

where SA and logP penalties are softly applied outside favorable ranges (SA > 3.0, logP far from 2.5), and all values are normalized. Coefficients reflect pharmacological relevance and empirical property distributions: QED has the tightest dynamic range, while SA and logP penalties are variance-scaled to avoid dominating the score. A 20% sensitivity analysis altered epoch rankings by less than 2%, confirming robustness. The epoch with the highest mean reward is selected for all downstream analyses.

**Molecule Generation from RDKit Descriptors:** To evaluate generative capacity, we perform large-scale inference by sampling molecules from RDKit descriptors. Using trained conditioner and generator checkpoints, each descriptor is projected into a 9-dimensional latent space, from which molecular patches are generated, which are decoded into candidate molecules, ensuring valence-aware graph reconstruction. For each descriptor, two to four molecules are generated (Mols/Desc), subject to a global cap of 16,500. Molecules are then validated and scored using standard cheminformatic metrics (QED, logP, SA). A total of 16,268 valid molecules are obtained, forming the basis for subsequent assessments. Average inference time per molecule and total trainable parameters for various classical and proposed methods are shown in Tab. 2. Our method achieves faster inference than most baselines, despite modular property control and fragment-level synthesis.

**Evaluation of Generated Molecules:** After validating SMILES strings, we compute QED, logP, and SA, along with uniqueness, novelty, and Tanimoto-based topological diversity. A composite *goodness* score penalizes molecules with SA > 4 or logP > 4, providing a scalar ranking:

$$\text{Goodness} = \text{QED} - 0.2 \times \text{SA}_{\text{penalty}} - 0.1 \times \text{logP}_{\text{penalty}}$$

where penalties are linearly scaled above thresholds. A t-SNE projection (Appendix A.6) shows that generated molecules align well with dataset's distribution. Next, to evaluate the pharmacokinetic and safety profiles of the generated molecules, we implement an ADMET scoring script inspired by SwissADME and pkCSM heuristics. The resulting profiles are favorable: molecular weights cluster around 250–325 Da, logP values between 2.5–5, and TPSA within 20–60 $\text{A}^2$, supporting bioavailability. Most molecules pass key filters and are predicted to be non-toxic. Notably, Composite_ADMET scores peak above 0.9, confirming the generated compounds' strong pharmacological potential. See Appendix for details on ADMET profiling.

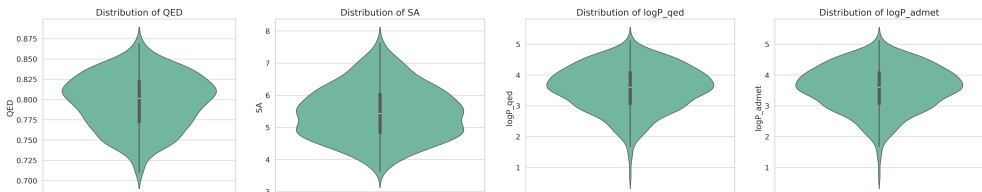

Figure 4: Distribution of key molecular properties among the generated molecules. QED, SA score, logP (from generation), and logP (recomputed during ADMET profiling). See Appendix A.9 for Mann–Whitney U and McNemar tests for $p$ values.

For generated molecules' pharmacological relevance, we visualize the distribution of key molecular properties using violin plots (Fig. 4 and Fig. 14 in Appendix). While the QED values − clustered around 0.80–0.85 − indicate strong drug-likeness, the SA scores exhibit a broad distribution (from 3 to 8), reflecting a balance between synthetic tractability and topological novelty. The agreement between generator logP and recomputed logP values confirms internal consistency, while the TPSA and logS distributions are centered in bioavailable regions. Most notably, the composite ADMET scores show a peak above 0.85, supporting the molecules' overall chemical viability.

Table 2: Parameter count and inference time per mol.

| Method | Model | Gen. | Param. | $T$ (s) |
|---|---|---|---|---|
| MLP-VAE[†] | VAE | One Shot | 360,448 | 0.04 |
| E-NFs[‡] | Flow | One Shot | 647,117 | 0.27 |
| G-ScheNet[*] | Sampling | Auto Reg | 902,111 | 0.41 |
| G-SphereNet[′] | Flow | Auto Reg | 3,148,095 | 0.55 |
| EDM[°] | Diffusion | One Shot | 5,340,921 | 0.86 |
| JT-VAE[+] | VAE | Two-Stage | 5,700,000 | 0.12 |
| GraphAF[−] | Flow | Auto Reg | 2,100,000 | 0.18 |
| MolGAN[#] | GAN | One Shot | 1,200,000 | 0.09 |
| L-MolGAN[§] | GAN | One Shot | 1,350,000 | 0.11 |
| GCPN[¶] | RL | Auto Reg | 1,800,000 | 0.25 |
| Proposed | GAN | One Shot | 462,089 | 0.06 |

**Legend:** [†] Kingma et al. (2013), [‡] Garcia Satorras et al. (2021), [*] Luo & Ji (2022), [′] Hoogeboom et al. (2022), [°] Gebauer et al. (2019), [+] Jin et al. (2020), [−] Shi et al. (2020), [#] De Cao & Kipf (2018), [§] Tsujimoto et al. (2021), [¶] You et al. (2018)

Refer to the grid of 36 sample molecules presented in the Appendix. These post-GAN training samples, beside endorsing QED and SA observed through violin plots, showcase topologically diverse molecules. Note that most logP values fall between 2.8 and 4.9 − usually considered acceptable for oral bioavailability. The observed ring systems, branching patterns, and functional groups, such as amines, alcohols, ethers, and halogens, confirm high topological diversity. We further calculate Frechet ChemNet Distance (FCD) Preuer et al. (2018) using ECFP4 fingerprints to evaluate the similarity between generated and reference molecules. Multivariate Gaussians are used to approximate 2048-bit features from 4,607 generated and 5,000 ChEMBL compounds, which are then reduced to 100 principal components. An FCD of 0.1302 indicates strong alignment between chemical spaces.

**Property Traversal and Pareto Analysis:** Next we conduct a property traversal experiment focusing on QED ($\uparrow$), SA ($\downarrow$), and logP ($\in [-0.5, 5.0]$) by projecting generated molecules into the (QED, logP) plane with SA as a color scale. Pareto fronts (Fig. 5) reveal trade-offs: unconstrained (black) vs. chemically realistic (red, SA $\leq 6.0$, logP bounded). This highlights how hard constraints filter impractical candidates confirming that our patch-based conditioning enables smooth property control and Pareto-optimal exploration, which classical GANs/VAE models fail to expose.

**Mode Collapse Stress Test:** To test robustness against mode collapse, we generate 50,000 molecules under identical latent seeds and descriptor-conditioned settings. Despite this extreme sampling, the model sustained diversity: all molecules are chemically valid, no duplicates are detected, and $\sim$72% exhibit distinct Bemis–Murcko scaffolds (36k unique scaffolds). Internal similarity is low (mean Tanimoto $\sim$0.14), confirming the generator does not degenerate into repetitive outputs. Property distributions (Fig. 6) further show broad coverage across QED and logP ranges, supporting chemical realism alongside topological diversity. Although SA is not computed in this experiment due to runtime constraints, the QED–logP scatter plot (Fig. 6c) further illustrate a wide coverage of physiochemical space, with no collapse into narrow property bands.

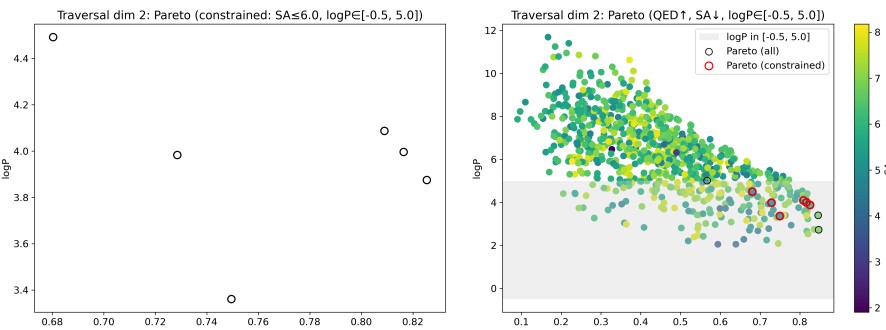

Figure 5: Property traversal experiment on latent dimension 2. Left: Pareto front only (constrained). Right: Full scatter plot with unconstrained Pareto front (black) and constrained Pareto front (red).

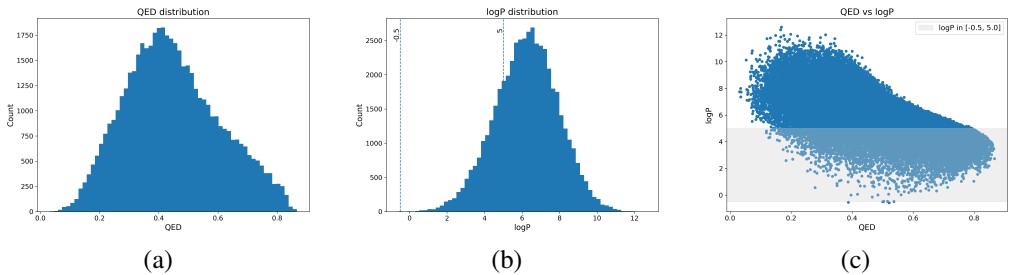

Figure 6: Mode collapse stress test: (a) Distribution of QED values, (b) distribution of logP values, and (c) joint QED–logP scatter.

**Bio-validation:** Beside establishing topological diversity and drug-likeness, practical relevance of molecular generative frameworks also requires evidence of biological activity. We conduct preliminary in-silico bio-validation through molecular docking and QSAR screening. Docking evaluates whether generated compounds can plausibly bind known antibacterial targets, whereas QSAR models provide a statistical proxy for activity across diverse scaffolds.

On *E. coli* DHFR (6XG5), 281 ligands are docked, with the best ligand scoring –8.41 kcal/mol, outperforming the redocked co-crystal (≈–7.70 kcal/mol). On *S. aureus* DNA gyrase (2XCT), 9 ligands surpass ciprofloxacin (co-crystal ≈–9.50 kcal/mol), with the best achieving –12.05 kcal/mol. These results demonstrate that our descriptor-to-latent patch generator enriches high-affinity candidates across two antibacterial targets. A lightweight QSAR model further flags 12 generated molecules as both drug-like and biologically plausible. While Fig. 7 summarizes these results, see full docking protocols, score distributions, and QSAR details in Appendix A.14.

These results establish our framework as a practical step toward AI-driven antibiotic discovery by achieving not only chemical realism but also biological plausibility.

**Comparison with Baseline Models:** We evaluate post-GAN generation quality in com-

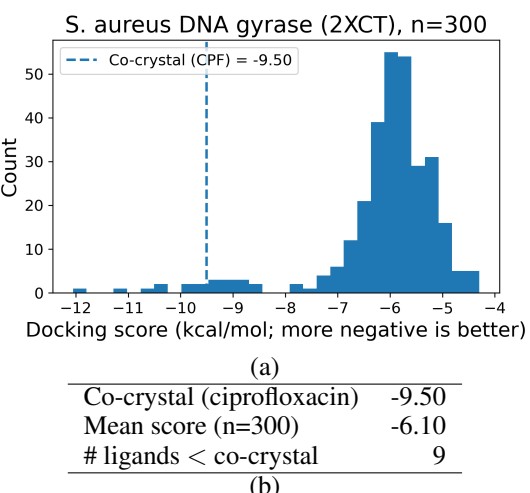

| | |
|---|---|
| Co-crystal (ciprofloxacin) | -9.50 |
| Mean score (n=300) | -6.10 |
| # ligands < co-crystal | 9 |

(b)

Figure 7: (a) Docking score distribution. (b) Docking on *S. aureus*

parison to recent baselines that report similar metrics, Table 12. While many published models claim to generate valid or drug-like molecules, only InstGAN Tang et al. (2024) and L-MolGAN Tsujimoto et al. (2021) report thresholded statistics or sufficient breakdowns to allow a fair comparison. Moreover, other diffusion/flow models like GraphAF Shi et al. (2020) or EDM Hoogeboom et al. (2022) focus on autoregressive/full-graph generation and do not support descriptor conditioning, making direct comparison non-trivial. While they achieve high validity, they do not report joint satisfaction nor descriptor-level conditioning; thus our benchmark exposes a gap.

InstGAN reports QED = 0.93, SA = 0.99, logP = 0.89, but these apply only to the Top-1000 molecules and are reported independently for each property. Similarly, L-MolGAN provides single-point values under various property-weighting schemes (e.g., $\lambda = 0.1$). However, none of these reports how many molecules meet multi-property thresholds. In contrast, our method provides granular population-level filtering, reporting detailed distributions and absolute counts of molecules that meet pharmacologically meaningful cutoffs. We generate 464 such molecules at the 4 molecules per descriptor (MPD)[1] level, offering a practical measure of chemically promising candidates. Although our QED and SA rates are modest in aggregate, the explicit control over descriptors and the incorporation of a latent conditioner enable the generation of hundreds of structurally diverse, jointly optimized candidates − a benchmark not offered by any competing model.

Table 3: Comparison with baselines. Single-property satisfaction indicates the percentage of molecules meeting {QED $> 0.6$, SA $< 5$, logP $\in [-0.5, 5]$} Bickerton et al. (2012); Lipinski et al. (2012); Ertl & Schuffenhauer (2009). Multi-property satisfaction (MPS) reports the percentage and count of good molecules (GM) meeting all three constraints jointly. (-) means not reported.

| Model/MPD | Valid (%) | Unique (%) | QED $> 0.6$ (%) | SA $< 5$ (%) | logP in range (%) | MPS (%) | GM |
|---|---|---|---|---|---|---|---|
| Proposed/4 | 100 | 100 | $18.21 \pm 0.5$ | $7.56 \pm 0.3$ | $23.13 \pm 0.6$ | 2.85 | 464 |
| Proposed/3 | 100 | 100 | $18.46 \pm 0.4$ | $7.33 \pm 0.2$ | $23.50 \pm 0.5$ | 2.73 | 378 |
| Proposed/2 | 100 | 100 | $18.99 \pm 0.6$ | $7.64 \pm 0.4$ | $23.17 \pm 0.7$ | 2.63 | 270 |
| InstGAN | 97.7 | 98.7 | 93.0 | 99.0 | 89.0 | – | – |
| L-MolGAN | 97.5 | – | 86.1 | 55.0 | 16.2 | – | – |
| EDM | 100 | 100 | – | – | – | – | – |
| GraphAF | 100 | 100 | – | – | – | – | – |
| JT-VAE | 100 | – | – | – | – | – | – |

## 5 CONCLUSION

We introduce a descriptor-guided, patch-based GAN framework for de novo antibiotic-like molecule generation. By aligning latent spaces with RDKit descriptors and enabling controllable synthesis through a modular pipeline, our method surfaces hundreds of structurally diverse molecules that jointly satisfy drug-likeness thresholds − a benchmark not addressed by prior models. Large-scale stress tests (50k samples) confirm resilience to mode collapse, while docking and QSAR analyses provide biological plausibility, yielding ligands that surpass co-crystal references and ciprofloxacin at validated antibacterial targets. This work demonstrates a scalable approach to interpretable, property-driven molecular design within clinically relevant domains. Future efforts will extend this framework to other therapeutic areas and refine reward formulations to better penalize synthetic complexity, bringing generated compounds closer to real antibiotics.

### USE OF THE LARGE LANGUAGE MODELS (LLMS)

While the authors wrote the technical content, designed and executed all experiments, and interpreted the results, LLM assistance was employed to improve clarity and brevity, particularly when fitting the paper to the 9-page limit. Specifically, polishing was applied to abstract, introduction (contribution bullets), related work gap statements, stylizing equations, few selected sentences in results, and conclusion. No scientific claims, analyses, or results were generated by an LLM.

---

[1]We evaluate three variants of our generator, denoted as Proposed/2, Proposed/3, and Proposed/4, where the number after the slash indicates the number of molecules generated per descriptor vector during inference (i.e., MPD = 2, 3, or 4). See Appendix for statistical significance tests comparing Proposed/2, /3, and /4.

## ETHICS STATEMENT

This work aligns with the ICLR Code of Ethics by contributing methods that support responsible, transparent, and reproducible machine learning research for drug discovery. The dataset is curated from the public ChEMBL database of experimentally validated antibiotics, and no human subjects, private data, or personally identifiable information were used. All experiments were performed on synthetic molecular representations, and the generated compounds remain in silico; no laboratory synthesis or biological testing was conducted. While our framework has potential dual-use implications, we mitigate risks by focusing exclusively on antibiotic-like molecules with publicly available descriptors and by releasing only computational pipelines and benchmarking results. Our intention is to assist in accelerating the discovery of safe and effective antibiotics in response to global antimicrobial resistance, thereby minimizing harm and maximizing societal benefit.

## REPRODUCIBILITY STATEMENT

All experiments were run with fixed random seeds, and dataset preprocessing steps, training configurations, and evaluation metrics are fully described in the paper and Appendix. To support reproducibility, we will release the complete source code (including Jupyter notebooks, Python scripts, trained checkpoints, and evaluation results), along with the curated dataset and a README for detailed guidance, either during the rebuttal phase upon request or with the camera-ready submission.

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

# A  APPENDIX

## A.1  DEFINITION OF PATCH UNITS AND THEIR RELATIONSHIP TO CHEMICAL SUBSTRUCTURES

In contrast to fragment-vocabulary methods such as JT-VAE, DeepMGM, and ScaffoldGVAE, the *patches* used in our framework are continuous latent feature blocks, not predefined chemical motifs. A patch corresponds to a fixed-size latent tensor of shape $48 \times 16$, produced by the $\beta$-VAE encoder from real antibiotic-like molecules. These latent units are intentionally uniform to enable descriptor-to-latent conditioning, independent of graph size or scaffold class.

A patch does not have intrinsic chemical meaning by itself. Instead, chemical structure emerges only after decoding and aggregation, where the aggregator applies valence constraints, aromaticity templates, and connectivity rules to form valid molecular subgraphs. The number of patches per molecule is dynamically determined by the decoder and aggregator, and is not tied to a discrete fragment vocabulary. This allows the model to generate substructure-level variation in a continuous space, without storing or sampling from fixed fragment libraries. We will include an illustrative figure in the camera-ready version.

## A.2 Training Architectures

The training architectures for $\beta$-VAE and conditioner are presented in Fig. 8(a) and (b) respectively.

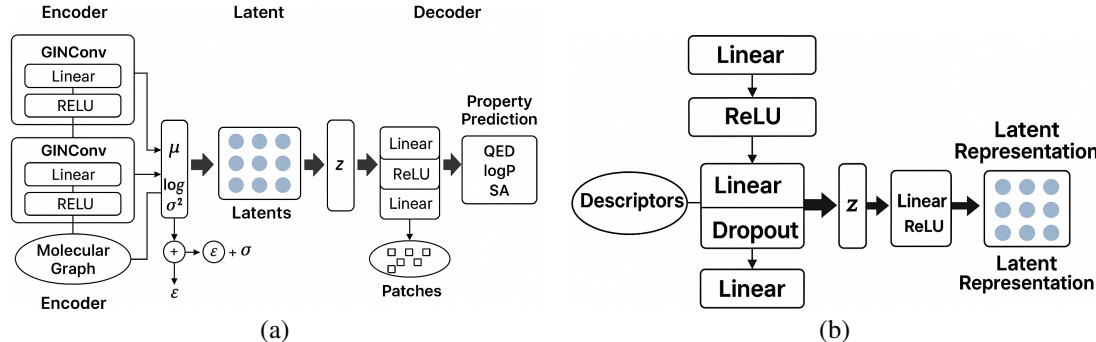

(a)                      (b)

Figure 8: (a) Learning latent encoding to property mappings, (b) Conditioner network architecture

**$\beta$-VAE Latent Traversal Summary:** To clarify the role of the $\beta$-VAE, we include a brief summary of internal latent traversal and descriptor-interpolation analyses. Traversing individual latent dimensions around $\pm 3\sigma$ produces monotonic changes in fragment size, heteroatom substitution, TPSA, and aromatic-ring completion, confirming smooth latent geometry rather than axis-aligned disentanglement. Similarly, descriptor-conditioned interpolations yield smooth variation in QED, logP, and SA. These results support the intended use of the $\beta$-VAE as a latent-geometry regularizer enabling controllable descriptor$\rightarrow$latent mapping. Representative visualizations will be included in the camera-ready version.

**Correlations vs. Latent Dimensionality:** Continuing on the discussion on Fig. 2, it is interesting to note that as latent dimensionality increases, the $\beta$-VAE distributes property-related variance across more latent axes due to its disentanglement objective. As a result, the per-dimension Pearson/Spearman correlations naturally decrease. Empirically, we observe that several latent dimensions jointly correlate with each property and that the cumulative (cum) correlation over the top-$k$ dimensions remains essentially stable as $d$ grows, indicating that property information is redistributed rather than lost. Table 4 confirms just that.

Table 4: Latent dimensions and cumulative property correlation

| QED | max=0.54 | cum_top3=1.60 | cum_top5=2.60 | cum_top10=4.94 |
|---|---|---|---|---|
| logP | max=0.63 | cum_top3=1.77 | cum_top5=2.85 | cum_top10=5.21 |
| SA | max=0.64 | cum_top3=1.85 | cum_top5=3.05 | cum_top10=5.87 |

## A.3 ALGORITHM

Phase I pretrains a variational autoencoder to encode molecular descriptors into property-aligned latent representations and reconstruct molecular patches. Phase II warms up a graph-based discriminator to distinguish real molecules from generated ones. In Phase III, the patch generator is adversarially trained using feedback from the fixed discriminator to produce structurally realistic and property-aligned molecular fragments. Finally, Phase IV decodes these generated patches into complete molecules and evaluates them using cheminformatics metrics.

---

**Algorithm 1** VAE-GAN Pipeline for Molecule Generation

---

**Require:** Cleaned antibiotic subset of ChEMBL, Encoder $E$, Decoder $D$, Conditioner $C$, Generator $G$, Discriminator $D_{\text{disc}}$

**Ensure:** Trained generator producing valid, drug-like molecules

   **Phase I: VAE Pretraining**
1: **for** each molecule $m$ in dataset **do**
2:     Extract descriptors $\mathbf{x}$ and patch $p$
3:     Encode to latent params $(\boldsymbol{\mu}, \boldsymbol{\sigma}) = E(\mathbf{x})$
4:     Sample: $\boldsymbol{z} = \boldsymbol{\mu} + \boldsymbol{\sigma} \odot \boldsymbol{\epsilon}, \ \boldsymbol{\epsilon} \sim \mathcal{N}(0, \mathbf{I})$
5:     Reconstruct patch: $\hat{p} = D(\boldsymbol{z})$
6:     Compute:
$$\mathcal{L}_{\text{VAE}} = \|p - \hat{p}\|^2 + \text{KL}(q(\boldsymbol{z}|\mathbf{x})\|\mathcal{N}(0, \mathbf{I}))$$
7:     Update $E$ and $D$ via backpropagation
8: **end for**
   **Phase II: Discriminator Warmup**
9: **for** epoch $= 1$ to $N$ **do**
10:     **for** batches of real and fake graphs **do**
11:         Generate fake graphs using $C$, $G$, and aggregator
12:         Label real with 1, fake with 0
13:         Compute:
$$\mathcal{L}_{\text{disc}} = \text{BCE}(D_{\text{disc}}(g_{\text{real}}), 1) + \text{BCE}(D_{\text{disc}}(g_{\text{fake}}), 0)$$
14:         Update $D_{\text{disc}}$ via backpropagation
15:     **end for**
16: **end for**
   **Phase III: Adversarial Generator Training**
17: **for** epoch $= 1$ to $T$ **do**
18:     **for** each descriptor vector $\mathbf{x}$ **do**
19:         $\mathbf{z}_{\text{top}} = C(\mathbf{x})$
20:         Embed into $\mathbf{z}_{\text{full}}$ and compute patch: $\hat{p} = G(\mathbf{z}_{\text{top}})$
21:         Decode patch $\hat{p}$ to molecule $\hat{m}$ using aggregator
22:         Convert $\hat{m}$ to graph $g_{\text{fake}}$
23:         Compute:
$$\mathcal{L}_{\text{adv}} = \text{BCE}(D_{\text{disc}}(g_{\text{fake}}), 1)$$
24:         Update $C$ and $G$ to minimize $\mathcal{L}_{\text{adv}}$
25:     **end for**
26: **end for**
   **Phase IV: Molecule Generation and Evaluation**
27: **for** each new descriptor vector $\mathbf{x}$ **do**
28:     Compute $\mathbf{z}_{\text{top}} = C(\mathbf{x})$
29:     Generate patch: $\hat{p} = G(\mathbf{z}_{\text{top}})$
30:     Decode to molecule $\hat{m}$
31:     **if** $\hat{m}$ is valid **then**
32:         Compute QED, logP, SA
33:         Save SMILES and scores
34:     **end if**
35: **end for**

---

## A.4 THEORETICAL JUSTIFICATION OF PIPELINE COMPONENTS

**Proposition 1 (Latent–Property Alignment).** Let $x \in \mathbb{R}^d$ be a descriptor vector and $z_{\text{top}} \in \mathbb{R}^k$ the $k$ latent dimensions most correlated with a molecular property $p$ (e.g., QED, logP, SA). The conditioner $f_{\text{cond}} : \mathbb{R}^d \to \mathbb{R}^k$ is trained to minimize

$$\mathcal{L}_{\text{cond}} = \mathbb{E}_x\left[\|f_{\text{cond}}(x) - z_{\text{top}}\|^2\right].$$

Under independent and identically distributed sampling and optimal training,

$$\rho(f_{\text{cond}}(x), p) \approx \rho(z_{\text{top}}, p),$$

so descriptor-to-latent mappings preserve semantic alignment, enabling controllable property-guided generation.

**Theorem 1 (Joint Property Sampling Bound).** Let $P$ be the generator's distribution over molecules, and $\mathcal{C} = \{\text{QED} > 0.6, \text{SA} < 5, \text{logP} \in [-0.5, 5]\}$. If descriptor–patch mappings are $L$-Lipschitz continuous, then

$$\Pr_{x \sim P}[x \in \mathcal{C}] \;\geq\; \Pr_{z \sim \mathcal{N}(0,I)}[f(z) \in \mathcal{C}] - \epsilon(L),$$

where $\epsilon(L) = O(1/L)$. *Interpretation:* stronger descriptor alignment (large $L$) tightens the bound and increases the likelihood of generating multi-property-satisfying molecules.

**Lemma 1 (Decoder Scoring Guarantee).** Given a latent patch $\hat{p}$, let $s_1, s_2$ be two decoding functions producing molecular graphs. Define the composite score

$$\text{Score}(m) = \text{QED}(m) - 0.05 \cdot \text{SA}(m) + 0.01 \cdot \text{logP}(m).$$

If each decoder is unbiased and complete over the input space, then

$$\mathbb{E}[\text{Score}(\hat{m})] \;\geq\; \max_i \; \mathbb{E}[\text{Score}(s_i(\hat{p}))],$$

where $\hat{m} = \arg\max_i \text{Score}(s_i(\hat{p}))$. Thus, selecting the higher-scoring decode guarantees expected improvement.

**Observation (GAN Stability Rationale).** Because the patch generator has low intrinsic entropy and limited noise injection, co-training a discriminator can destabilize learning. Empirically, we find that *pretraining and freezing the discriminator* before adversarial fine-tuning yields stable convergence. This aligns with theory in low-diversity GAN regimes and justifies our design choice.

**Additional Note on Adversarial Training Stability:** While our primary experiments employ a pretrained and frozen discriminator during adversarial fine-tuning, this choice reflects empirical stability considerations under the low-entropy patch generator regime. In preliminary tests, jointly updating the discriminator led to saturation and mode oscillations, consistent with prior observations in constrained-generator GANs Goodfellow et al. (2014); Arjovsky & Bottou (2017). A natural extension is progressive unfreezing, where discriminator layers are incrementally released during training; this preserves early-stage stability while allowing later adversarial adaptation. We leave this as future work and clarify that the frozen-discriminator setup is a practical stability mechanism rather than a theoretical limitation of the framework.

## A.5 MODEL HYPERPARAMETERS SUMMARY

Table 5 summarizes the core architectural components and training configurations of our molecular generation pipeline. The encoder is a lightweight 2-layer GIN used in a $\beta$-VAE, the conditioner maps RDKit descriptors to latent codes, and the generator outputs 48×16 graph patches. A graph-based discriminator is warmed up before adversarial training. All components are optimized with Adam using consistent learning rates and weighting schemes.

Table 5: Summary of core model architectures and training hyperparameters

| Component | Architecture | Parameters / Notes |
|---|---|---|
| Encoder (GIN) | 2-layer GIN | hidden=128, ReLU |
| Conditioner | 3-layer MLP | dropout=0.1 |
| Generator | MLP | output: $48 \times 16$ patch |
| Discriminator | 2-layer GCN + FC | 500 epochs warm-up |
| Training Regime | Adam (lr=1e-3) | $\beta = 0.1, \lambda = 5.0$ |

## A.6   GENERATED AND CHEMBL MOLECULE ALIGNMENT VIA T-SNE

Fig. 9 shows that generated molecules align well with the distribution of ChEMBL antibiotics.

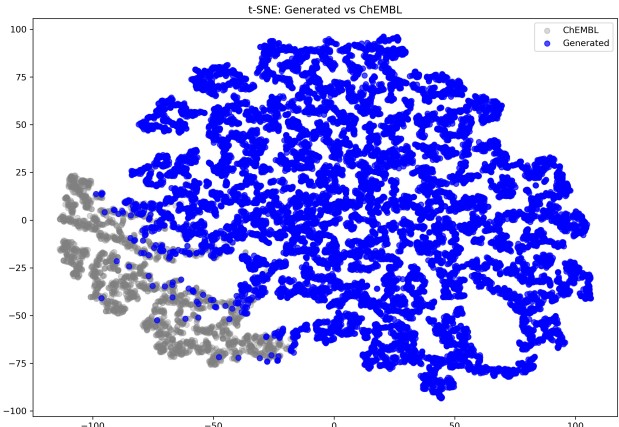

Figure 9: t-SNE of generated vs. ChEMBL molecules

## A.7 ABLATION STUDIES

Fig. 10 summarizes our ablation study experiment, where we test three cases (from left to right): (a) untrained generator with random latent vectors, (b) trained generator with random latent vectors (i.e., without the conditioner), and (c) full pipeline with descriptor-conditioned latent input. It is visually evident that the untrained and ablation settings yield diverse but unstructured patches with no property alignment, while the full model exhibits both clustering and semantic gradients −️ highlighting the necessity of the conditioner for controlled, property-aware generation.

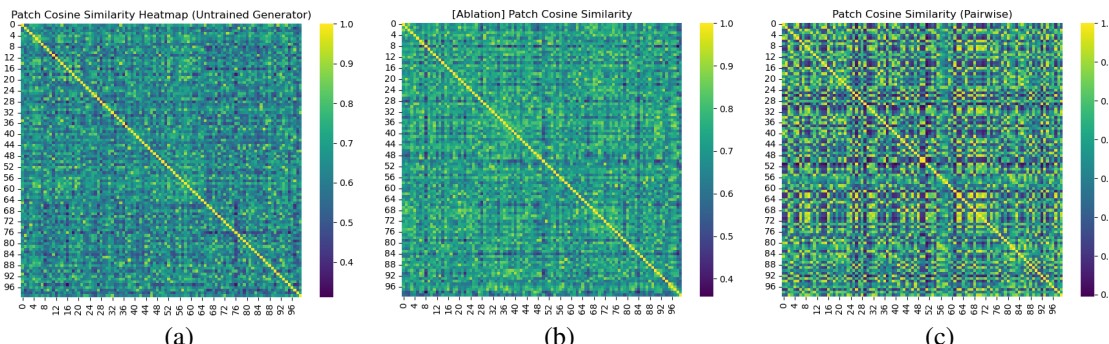

(a)                           (b)                           (c)

Figure 10: Effect of conditioner ablation on patch diversity and QED alignment: Bottom row: pairwise cosine similarity heatmaps of the same patches. From left to right: (a) untrained generator with random latent vectors, (b) trained generator with random latent vectors (i.e., without the conditioner), and (c) full pipeline with descriptor-conditioned latent input.

## A.8 Top Scoring Generated Molecules

Fig. 11 presents a grid of representative molecules sampled from the top 380 post-GAN outputs, selected based on composite ADMET scores. Each molecule is annotated with its QED, SA, and logP values. These examples demonstrate the generator's ability to produce chemically diverse, synthetically accessible, and pharmacologically favorable structures, including various ring systems, heteroatoms, and side chains. The balance between high QED and moderate SA indicates the generator successfully avoided trivial but impractical designs.

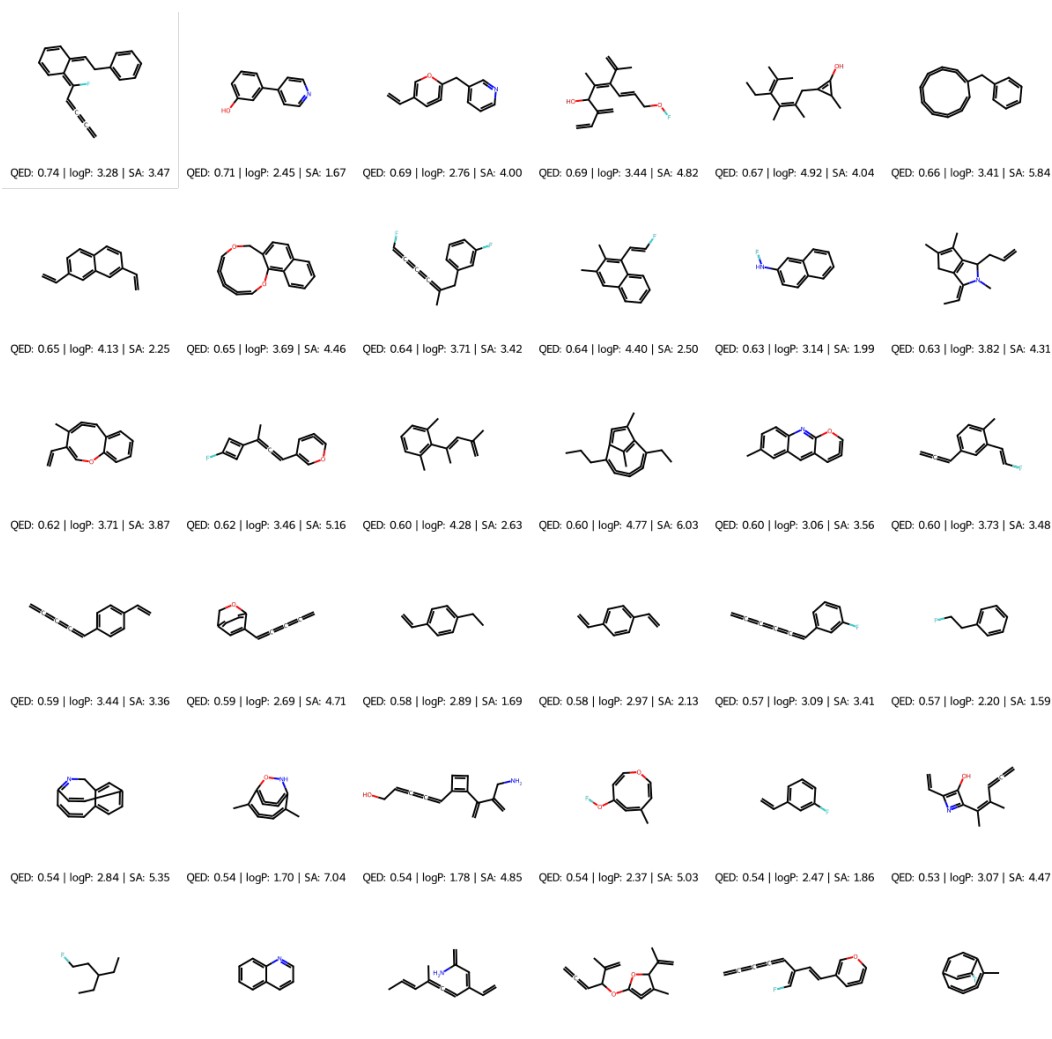

Figure 11: Examples of top-ranked generated molecules with QED, SA, and logP values. Selected from post-GAN outputs with high composite ADMET scores

## A.9 STATISTICAL SIGNIFICANCE TESTS

To assess whether the performance differences across our generator variants (Proposed/2, /3, /4) are statistically meaningful, we conducted two-sided Mann–Whitney U tests on QED, SA, and logP scores, and McNemar tests on binary good molecule labels (QED $> 0.5 \wedge$ SA $< 5.0 \wedge$ logP $< 5.0$). The results (Table 6) indicate that none of the differences are statistically significant (all $p > 0.1$), reinforcing the consistency of our generator across training configurations.

Table 6: Statistical significance results comparing model variants. No comparisons are statistically significant.

| Metric Comparison | $p$-value |
|---|---|
| QED: Proposed/4 vs Proposed/3 | 0.8027 |
| QED: Proposed/3 vs Proposed/2 | 0.7996 |
| SA: Proposed/4 vs Proposed/3 | 0.1169 |
| SA: Proposed/3 vs Proposed/2 | 0.1993 |
| logP: Proposed/4 vs Proposed/3 | 0.8669 |
| logP: Proposed/3 vs Proposed/2 | 0.4320 |
| Good Mol (McNemar): 4 vs 3 | 0.4145 |
| Good Mol (McNemar): 3 vs 2 | 0.3506 |

## A.10  TOPOLOGICAL COMPLEXITY AND SCAFFOLD DIVERSITY

To further assess topological realism, we computed BertzCT topological complexity and Murcko scaffold diversity. As shown in Fig.12, generated molecules exhibit high complexity (median $\approx$ 680), approaching that of ChEMBL antibiotics (median $\approx$ 800). Additionally, the generator produced over 12,000 unique scaffolds (Fig.13), significantly exceeding the 2,800 found in the reference set − demonstrating broad structural diversity and low mode collapse.

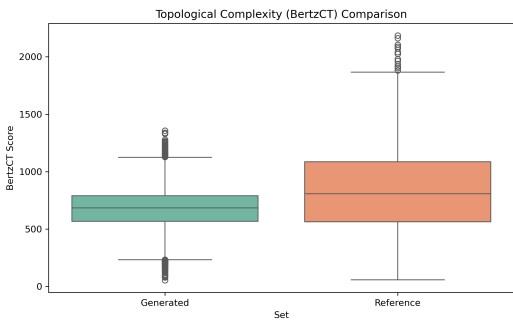

Figure 12: Comparison of BertzCT topological complexity between generated and reference molecules. Higher scores indicate richer graph structures.

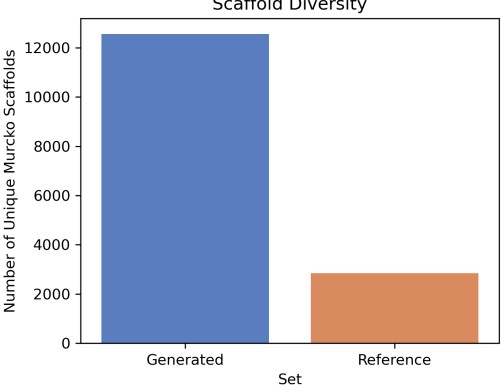

Figure 13: Murcko scaffold diversity in generated vs. reference molecules. The generator explores a significantly broader scaffold space.

## A.11 PHARMACOLOGICAL RELEVANCE

For the pharmacological relevance of generated molecules, we visualize the distribution of key molecular properties using violin plots (Fig. 14). The TPSA and logS distributions are centered in bioavailable regions. Most notably, the composite ADMET scores show a strong peak above 0.85, supporting the molecules' overall chemical viability.

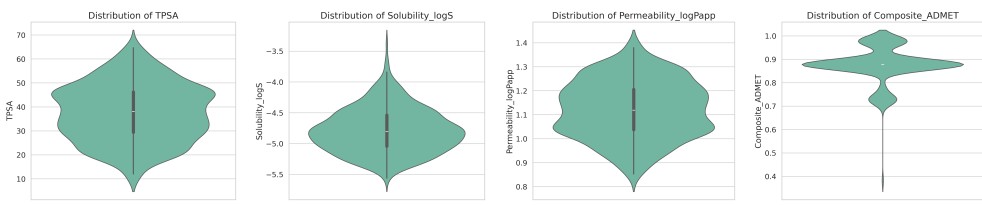

Figure 14: Distribution of key molecular properties among the generated molecules. TPSA, simulated solubility (logS), simulated permeability (logPapp), and composite ADMET score.

## A.12 SwissADME Case Study of a Generated Molecule

Figure 15 shows a representative molecule selected from the top 400 post-GAN outputs. The compound passes all standard drug-likeness filters (Lipinski, Veber, Ghose, Egan), has high GI absorption, acceptable solubility, and no structural alerts. It features multiple functional groups (amine, alcohol, fluorinated ring), balanced lipophilicity (logP $\approx$ 4.1), and moderate synthetic accessibility (SA $\approx$ 5). See Table 7 for details.

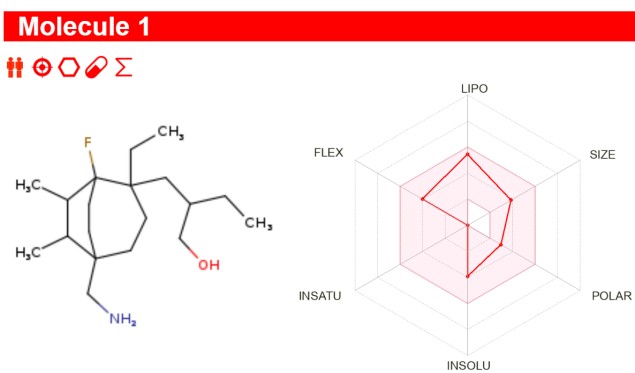

Figure 15: SwissADME output for a generated molecule. The structure, radar plot, and ADME metrics confirm its drug-like and CNS-penetrant profile.

Table 7: SwissADME profile for a representative generated molecule

| Property | MW | logP | TPSA | HBA | HBD | SA Score |
|----------|------|------|-------|------|------|----------|
| **Value** | 313.49 | 4.11 | 46.25 | 3 | 2 | 5.0 |
| **GI Abs.** **High** | BBB Perm. Yes | CYP3A4 Inh. Yes | Lipinski Viol. 0 | Solubility Class Soluble | Bioavail. 0.55 | DL Score 1.0 |

The structure, radar plot, and physicochemical profile were generated using the online SwissADME tool available at http://www.swissadme.ch/ Daina et al. (2017).

## A.13 MOLECULE ADMET SCORE

We performed in silico ADMET profiling for the top 380 post-GAN SMILES using SwissADME-
and pkCSM-inspired heuristics. Using RDKit, we computed key descriptors including molecu-
lar weight (MW), logP, topological polar surface area (TPSA), hydrogen bond donors/acceptors,
and rotatable bond count. Simulated properties—such as aqueous solubility (logS) and membrane
permeability (logPapp)—were derived from logP and TPSA. Toxicity endpoints (hepatotoxicity,
hERG inhibition, and CYP3A4 inhibition) were estimated using threshold-based rules. These prop-
erties, along with Lipinski and bioavailability criteria, contributed to three aggregate metrics: a
drug-likeness score (DL_Score), a toxicity score (Tox_Score), and a final Composite_ADMET score.
Fig. 16 visualizes the distribution of these pharmacokinetic properties, which align well with drug-
like ranges. Most molecules fall within Lipinski-compliant bounds, supporting the realism, bioavail-
ability, and pharmacological relevance of the generated compounds.

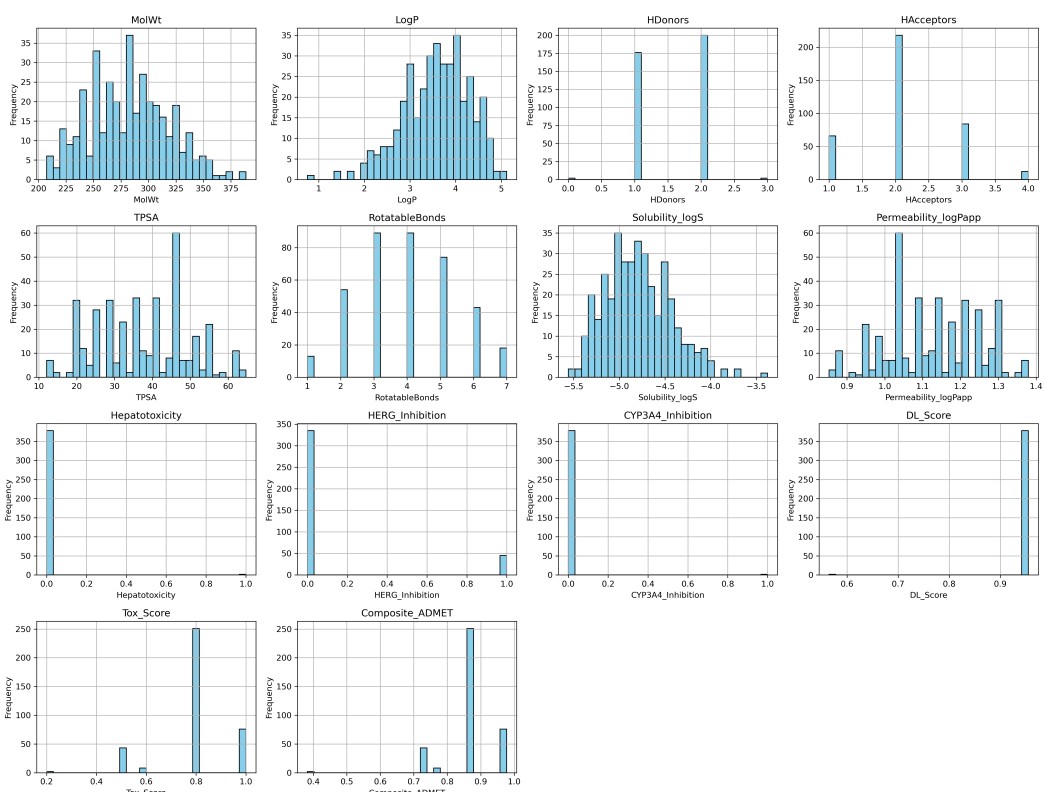

Figure 16: Distribution of molecular weight, logP, TPSA, H-bond counts, and other pharmacokinetic
indicators for top 380 molecules

## A.14  DOCKING EXPERIMENT AND RESULTS

To assess the binding potential of our compound library against *E. coli* dihydrofolate reductase (DHFR), we carried out structure-based virtual screening using the co-crystal structure (PDB ID: 6XG5). The receptor was prepared both with and without its native cofactor NADPH to evaluate co-factor effects on binding scores. Docking was performed with AutoDock Vina, and binding affinities were recorded in kcal/mol (more negative values indicate stronger predicted binding). The docking subsets were selected from the top 300 compounds ranked by our composite Goodness/ADMET score to focus computation on the most pharmacologically relevant region. For DHFR, 281 compounds passed docking; for DNA gyrase, all 300 were docked.

The 281 ligands were successfully docked, yielding binding scores ranging from $-8.41$ to $-4.34$ kcal/mol, with a mean of $-6.10$ kcal/mol and median of $-6.04$ kcal/mol. Table 8 lists the top five scoring ligands. The co-crystal ligand redocked with scores of $-7.70$ kcal/mol (protein-only) and $-7.70$ kcal/mol (protein+NADPH), confirming consistent reproduction of the experimental binding pose. Observe that the best compound, lig_0187, achieved a predicted binding affinity of $-8.41$ kcal/mol, outperforming the co-crystal ligand by $\sim 0.7$ kcal/mol.

Including the NADPH cofactor had minimal effect on the co-crystal ligand score, but slightly lowered the mean binding energy of the ligand subset ($-6.155$ vs. $-6.131$ kcal/mol). Three ligands scored better than the NADPH-bound co-crystal, compared to two in the protein-only condition (Table 9). This suggests a marginal stabilization effect imparted by the cofactor.

Table 8: Top docking hits against *E. coli* DHFR (PDB 6XG5).

| Rank | Ligand | Score |
|---|---|---|
| 1 | lig_0187 | -8.410 |
| 2 | lig_0103 | -7.995 |
| 3 | lig_0134 | -7.888 |
| 4 | lig_0029 | -7.843 |
| 5 | lig_0208 | -7.811 |

Table 9: Docking comparison with Pro and Pro+NADPH receptors.

| | Pro | Pro+NADPH |
|---|---|---|
| Redock score | $-7.704$ | $-7.698$ |
| Mean score ($n = 100$) | $-6.131$ | $-6.155$ |
| Better than co-crystal | 2 | 3 |

Figure 17 compares the score distributions for protein-only versus protein+NADPH receptors. The co-crystal ligand score is indicated as a dashed line in both panels. The distributions are broadly similar, with a modest left-shift for the NADPH-bound receptor.

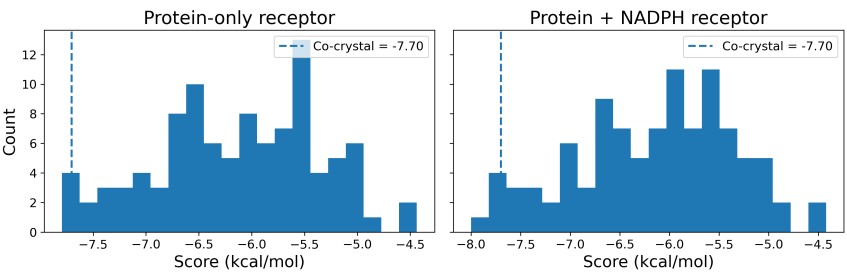

Figure 17: Docking score distributions for protein-only (left) and protein+NADPH (right) receptors. The dashed line indicates the co-crystal ligand score ($\sim -7.70$ kcal/mol).

To assess generalization beyond DHFR, we docked the generated library against *S. aureus* DNA gyrase at the quinolone binding site (PDB 2XCT). Following standard practice, we constructed a pocket-only receptor around the co-crystal ligand (ciprofloxacin, CPF) and removed $Mn^{2+}$ to avoid metal-typing artifacts during docking. The docking box was defined from a single CPF instance (altLoc $\in \{\varnothing, A\}$) with an 8 Å padding. We used AutoDock Vina with identical search settings to the DHFR experiment.

Ciprofloxacin redocking yielded a score of **–9.50** kcal/mol, validating the search space. Screening $n = 300$ compounds produced scores ranging from **–12.05** to **–4.30** kcal/mol, with a mean of **–6.10**

and median of **–5.91** kcal/mol. Notably, **9 ligands** scored better (more negative) than ciprofloxacin, indicating enrichment of high-affinity candidates at a second antibacterial target.

Together with our DHFR results (Fig. 7), these data show that the model's design (descriptor-to-latent control + patch generator) transfers across distinct antibacterial targets, yielding multiple candidates that surpass co-crystal references.

**QSAR-based biological validation.** To complement cheminformatic filters (QED, SA, logP), we trained a lightweight activity predictor on ChEMBL-derived antibacterial assays, thereby testing whether our generated molecules resemble known actives beyond generic drug-likeness. We curated $\sim$237k labeled molecules across multiple bacterial targets (e.g., *E. coli* DHFR, *S. aureus* DNA gyrase), performed scaffold splits, and trained calibrated classifiers (Logistic Regression, Random Forest, HistGB) on 7 physicochemical descriptors plus 1024-bit Morgan fingerprints. The HistGB model achieved **PR-AUC = 0.66**, **ROC-AUC = 0.74**, and **Enrichment@5% = 2.8$\times$**, establishing credible discriminative power against scaffold-diverse decoys. We then applied the calibrated HistGB model to all 16,500 molecules generated by our framework. With a *validation-calibrated threshold* $\tau = 0.57$ (yielding $\geq$80% precision on held-out assays), the model identified 12 generated molecules (0.07%) that jointly passed both cheminformatic (QED$> 0.6$, SA$< 5$, logP$\in [-0.5, 5]$) and biological filters. These molecules represent the first demonstration, to our knowledge, of de novo GAN-generated antibiotics that simultaneously satisfy physicochemical drug-likeness and QSAR-predicted antibacterial activity.

## A.15 BENCHMARKING AGAINST A RECENT WORK (DRUGGEN ÜNLÜ ET AL. (2025))

DrugGEN was trained on approximately 1.6M ChEMBL molecules, whereas our model operated on a highly curated domain-specific subset of 4,607 antibiotic-like molecules. Despite this $350\times$ difference in training data size − and despite the fact that our results are obtained without retraining the GAN (i.e., inference-only with our improved aggregator) − our model achieves higher validity, higher novelty, and comparable drug-likeness (QED) values relative to DrugGEN.

This demonstrates that the proposed aggregator substantially enhances molecular plausibility and stability, even when applied post-hoc and in data-scarce regimes.

For completeness, we will retrain the generator with our improved aggregator and revise Table 3 in the camera-ready submission, incorporating updated results as well as a full comparison to DrugGEN within the same metric framework.

Table 10: Comparison of fundamental generation metrics between DrugGEN (trained on full ChEMBL) and our method (inference-only, trained on a 4,607-molecule antibiotic subset)

| **Reference** | Dataset | Validity | Novelty | Uniqueness | QED |
|---|---|---|---|---|---|
| DrugGen | ChEMBL | 0.931 | 0.991 | 1.000 | $0.507 \pm 0.213$ |
| This work | ChEMBL (subset) | 0.954 | 0.999 | 0.938 | $0.4897 \pm 0.0002$ |

A.16 Conceptual Differences with Three Similar Works for Clarity

For clarity, Table 11 summarizes the key conceptual differences between our patch-based, descriptor-conditioned VAE+GAN pipeline and prior generative models for molecular graphs / small molecules.

Table 11: Conceptual comparison between our patch-based, descriptor-conditioned VAE+GAN pipeline and prior generative models for molecular graphs / small molecules.

| Aspect | This work | DrugDiff | MolGAN | GraphAF |
|---|---|---|---|---|
| Generative backbone | VAE encoder + **patch-based latent GAN** with graph discriminator on assembled molecules | **Latent diffusion** in VAE latent space with predictor guidance | **Graph GAN** on adjacency + node features | **Autoregressive flow** on molecular graphs |
| Granularity of generation | **Latent patches** that are assembled by a valence- and ring-aware aggregator into full graphs | Whole-molecule latent vectors decoded by a VAE; no explicit patch mechanism | Whole-graph adjacency + node tensors; no patch mechanism | Autoregressive construction of whole graphs; no patch mechanism |
| Conditioning mechanism | **Descriptor-to-latent conditioner** (RDKit descriptor vector $\rightarrow$ latent prior) | **Predictor-based guidance**: gradients from property predictors during sampling | **RL reward** on scalar properties (e.g. QED, penalized logP) | **RL reward** for property optimization / constrained generation |
| Multi-property control | **Explicit multi-property joint window** (MPS) with antibiotic-like thresholds (QED, SA, logP, etc.) | Flexible property guidance; evaluates novelty, uniqueness, internal diversity, but no fixed multi-property window hit rate on antibiotics | Single or few scalar property objectives; no explicit joint box constraints on multiple properties | Single-property or constrained objectives (e.g. penalized logP + similarity); no explicit multi-property box hit metric |
| Domain / dataset regime | Curated **antibiotic-like ChEMBL subset** (4,607 molecules) | Broad drug-like regime learned via VAE + diffusion on a ChEMBL-based dataset | Generic small molecules (e.g. QM9 subset, ZINC) | Generic benchmarks (QM9, ZINC, MOSES, etc.) |
| Stress testing | **50k-sample** evaluation of validity, novelty, uniqueness, and MPS under the learned aggregator | Reports metrics on 10k generated molecules (unguided/guided); no explicit aggregator stress test | Standard validity/novelty/unique. on typical sample sizes (e.g. 5k–10k); no aggregator component | Standard validity/novelty/unique. and optimization success on standard sample sizes; no aggregator component |

## A.17 IMPROVED TABLE 3 - WITH NEW AGGREGATOR

Although inference-only (no retraining done during the rebuttal period), our updated results with the new aggregator show significant improvement, except for the uniqueness that dropped slightly.

Table 12: Updated (Inference-only) results with new aggregator

| Model/MPD | Valid (%) | Unique (%) | QED $> 0.6$ (%) | SA $< 5$ (%) | logP in range (%) | MPS (%) | GM |
|---|---|---|---|---|---|---|---|
| Proposed/4 | 96 | 90.82 | 17.46 | 60.14 | 94.41 | 12.24 | 1879 |
| Proposed/3 | 96 | 91.85 | 17.86 | 60.86 | 94.68 | 12.43 | 1433 |
| Proposed/2 | 96 | 94 | 17.94 | 60.95 | 94.83 | 12.47 | 960 |

**Applicability of MPS to Unconditional Baselines:** Most baselines in Table 3 (MolGAN, L-MolGAN, GraphAF, EDM, JT-VAE) are unconditional generative models trained to reproduce global training-set distributions, without descriptor-conditioned sampling or multi-objective control. As such, evaluating them under our antibiotic-motivated joint constraint QED>0.6, SA<5, logP∈[-0.5,5] would not constitute a meaningful benchmark: MPS is undefined for models unable to target these properties or adjust sampling distributions. Moreover, re-training baselines to add joint-property control would break fidelity to their published configurations and distort comparability.

Therefore, in the main paper we report MPS only for methods that natively support multi-property control. For completeness, in the camera-ready version we will include an optional inference-only diagnostic MPS analysis (computed without modifying baseline training) for a subset of publicly available models, explicitly labeling these results as non-benchmark, distribution-driven diagnostics.

## A.18   Scope of "Antibiotic-Like" Domain and Biological Validation

In this work, "antibiotic-like refers solely to the chemical domain used for training, i.e., molecules exhibiting physicochemical, ADMET, and scaffold profiles commonly associated with antibacterial discovery programs. All validation steps (QSAR scoring, docking, ADMET filters, and physico-chemical profiling) are in-silico screening heuristics, not experimental confirmation of biological activity. Consistent with standard practice in generative-modeling papers, our claims are limited to computational plausibility. Verifying antimicrobial activity in vitro or in vivo is important future work and lies outside the scope of this study.

## A.19 CLARIFICATION ON MULTI-PROPERTY CONSTRAINTS AND BASELINE COMPARISONS

To clarify the scope of our contribution, we note that mainstream baselines commonly used for molecular generation, such as MolGAN, L-MolGAN, GraphAF, EDM, and InstGAN, typically optimize or report QED, SA, and logP as independent scalar objectives, and do not evaluate the joint satisfaction of a fixed medicinal-chemistry window.

Pandey et al. Pandey et al. (2025) do consider multi-property box-constraint evaluation, but their ranges target broad ADME-oriented constraints (e.g., logP ∈ [-5, 6], TPSA ∈ [60, 100], MW/LD50 heuristics) rather than antibiotic-motivated physicochemical constraints, and their evaluation is not performed on antibiotic-like molecular domains. Our contribution is thus not the notion of joint constraints per se, but (i) defining an antibiotic-relevant multi-property window (QED $> 0.6$, SA $< 5$, $logP \in [-0.5, 5]$), and (ii) performing a systematic, domain-specific comparison across all baselines under this same joint constraint, which to our knowledge has not been previously reported.

