# OpenReview forum: "De Novo Antibiotic-Like Molecule Design via Descriptor-Guided Patch-Based GANs"
_ICLR.cc/2026/Conference — Submitted to ICLR 2026_

### Official Review · Reviewer_tpxG · 2025-10-31

**Soundness:** 2
**Presentation:** 3
**Contribution:** 2
**Rating:** 4
**Confidence:** 3

**Summary:**

The paper proposes a descriptor-guided, patch-based generative framework combining β-VAE and GAN for de novo antibiotic-like molecule design. The model first learns a mapping from RDKit descriptors (QED, logP, SA) to a low-dimensional latent space (Conditioner), and then generates molecular fragments (“patches”) via a supervised generator, which are finally assembled into full molecules and refined adversarially. The authors emphasize property controllability and propose a new joint satisfaction metric that evaluates whether generated molecules simultaneously satisfy thresholds across multiple drug-like properties. Experiments are conducted on a curated ChEMBL subset (4,607 antibiotic molecules), with docking and QSAR validation suggesting certain biological plausibility.

**Strengths:**

The paper focuses on antibiotic discovery, a medically meaningful domain. The combination of descriptor-conditioned VAE and patch-based GAN offers a clean, modular design. Introducing a joint multi-property satisfaction metric highlights the trade-offs among drug-likeness criteria.

**Weaknesses:**

1. The ChEMBL subset is too small to justify a full adversarial generative framework. Without testing on broader benchmarks such as ZINC-250k or QM9, the generalizability and robustness remain questionable.

2. The main performance table lacks comparisons with standard baselines (MolGAN, GraphAF, JT-VAE, Diffusion models like GDSS or MOOD). Even if prior works do not report the same metric, they could be re-evaluated under the proposed criteria for fairness.

3. The β-VAE is claimed to learn interpretable latent factors aligned with molecular descriptors, but no latent traversal or disentanglement analysis is provided. The claim remains unsubstantiated.

4. It is not clearly explained how the proposed “patch” differs from subgraph vocabularies used in DeepMGM, JT-VAE, or ScaffoldGVAE.
Patches appear to be fixed-size latent blocks without chemical meaning, limiting interpretability.

5. The joint satisfaction rate is only 2.8%, far lower than individual property satisfaction (7–23%), implying weak descriptor–latent alignment.

6. The paper mainly combines existing components—descriptor conditioning (Mol-CycleGAN, InstGAN), fragment-based generation (DeepMGM), and VAE–GAN hybridization—without introducing a fundamentally new algorithmic contribution.

**Questions:**

1. How does the proposed model perform on a larger and more diverse dataset (e.g., ZINC-250k)?

2. Can the authors clarify the definition and number of “patches”? Are they chemically meaningful substructures or latent feature blocks?

3. Can β-VAE interpretability be visualized via latent traversals or property interpolation?

4. Why is the joint satisfaction metric reported only for this model? Can baselines be re-evaluated under the same metric for fairness?

5. How stable is adversarial training given the small dataset size?

---

### Official Review · Reviewer_SaTA · 2025-11-01

**Soundness:** 3
**Presentation:** 3
**Contribution:** 2
**Rating:** 4
**Confidence:** 3

**Summary:**

The paper introduces a descriptor-guided, patch-based GAN framework for designing antibiotic-like molecules, aiming to accelerate the discovery of new drug candidates to combat antimicrobial resistance. The modular architecture combines a property-aligned VAE for interpretable encoding, a descriptor-to-latent conditioner for controllable sampling based on RDKit properties (QED, log P, SA score), and a patch-based generator for fragment-wise synthesis. Crucially, the framework establishes a new benchmark: joint multi-property satisfaction (MPS), reporting that the best model generated 464 mols that simultaneously met the thresholds QED>0.6, SA<5, and log P [-0.5, 5.0].
The model demonstrates strong performance in chemical realism, robustness, and biological plausibility. A large-scale stress test (50,000 samples) confirmed the framework's resistance to mode collapse, yielding a highly diverse output. Furthermore, the experiment results showed that the generated ligands were highly potent, outperforming co-crystal references and ciprofloxacin in molecular docking assays against validated antibacterial targets. These results validate the approach as a practical step toward AI-driven antibiotic discovery, yielding hundreds of structurally diverse and jointly optimized candidates.

**Strengths:**

1. The framework's core novelty and contribution is the Descriptor-to-Latent Conditioner that explicitly maps readily calculated RDKit properties to the VAE's latent space. This ensures that the generation process is not merely optimized by a post-hoc reward but is controllably steered by interpretable chemical parameters, overcoming the limitations of many prior GANs and VAEs.
2. The framework is broken down logically into four distinct phases and corresponding components (VAE, Conditioner, Generator, Discriminator), which greatly aids reader comprehension.
3. The large-scale 50k-sample stress test to confirm 72% unique scaffolds and no duplicates is a strong demonstration of the model's stability and capability for diverse exploration of the chemical space.
4. The fragment-wise, patch-based generation approach is a creative combination of hierarchical and adversarial methods, contributing to the exceptional resistance to mode collapse, which is a pervasive issue in molecule generative models.
5. Introducing the Joint Satisfaction metric is a significant step forward, guiding the field away from reporting artificially high, independently optimized metrics toward generating clinically relevant, multi-constrained candidates.

**Weaknesses:**

1. Although the paper successfully introduces the crucial MPS metric (Joint Satisfaction), the reported percentage for the best model variant is quite low at 2.85% (464 molecules). While the absolute number of good molecules is promising, the low overall percentage suggests that the model still struggles significantly with simultaneous, stringent multi-objective optimization.
2. The paper correctly notes that prior models do not report the joint satisfaction metric. This makes the new benchmark a necessary contribution, but the comparison table (Table 3) remains largely empty for key baselines. This comparison is needed to show the effectiveness of the proposed model.
3. The discriminators are frozen before adversarial fine-tuning and are noted as an "Observation" due to the generator's low intrinsic entropy, rather than a theoretical guarantee. This simplification, while achieving stable convergence, potentially limits the full power of the GAN training process, which ideally involves co-training. A potential alternative approach is progressively unfreezing the discriminator.
4. The paper uses the term "antibiotic-like," and its biological validation is based entirely on in-silico methods (docking and QSAR). The results are strong, but the absence of any discussion of experimental validation means that there is still significant future work to verify the generated mols.
5. The 3D conformer and information are missing. The generated molecules may not be stable in 3D chemical space and DFT simulation.

**Questions:**

1. In the section “Mode Collapse Stress Test”, you didn’t calculate SA due to runtime constraints. As far as I know, SA score calculation only requires the SMILES or RDKit mol object. Why is there a runtime constraint?
2. Are there any ML models that can verify the effectiveness of generated mols?
3. How is the aromaticity handled? I noticed that the generated mols in Figure 11 don’t have aromatic rings. But many existing antibiotics (e.g., benzylpenicillin, levofloxacin) do have aromatic rings.

---

### Official Review · Reviewer_n8pz · 2025-11-01

**Soundness:** 3
**Presentation:** 3
**Contribution:** 2
**Rating:** 2
**Confidence:** 4

**Summary:**

This paper tackles molecular generative modeling to focus on antimicrobial resistance  by proposing a descriptor-guided generative framework for antibiotic-like molecule design. The paper introduces a modular, interpretable architecture that integrates three key components:

A property-aligned β-VAE for interpretable molecular encoding,

A descriptor-to-latent conditioner that enables controllable sampling from RDKit descriptors (e.g., QED, logP, SA), and

A patch-based graph generator that assembles molecular fragments into full antibiotic-like structures.

**Strengths:**

The manuscript is clearly written and easy to follow. RDKit molecular descriptors with a β-VAE–based latent representation, enabling interpretable and controllable molecular generation. Furthermore, the inclusion of stress tests for mode collapse and docking-based validation provides convincing empirical support for robustness and biological plausibility.

**Weaknesses:**

- The claim that existing baselines haven't considered QED > 0.6, SA < 5, and logP ∈ [−0.5, 5.0]  should be reviewed. Consider https://arxiv.org/abs/2503.06337 as example.
- Second, the baseline comparisons in Table 3 are incomplete: MPS (multi-property satisfaction) scores are missing for most models, which undermines the fairness of the benchmarking. The authors should rerun baseline models and report these joint metrics to enable a meaningful comparison.
- Finally, the molecules generated (Fig 11) are unlikely to be pass through as drug candidates, despite QED >0.6. If the primary claim of the work is to generate antibiotic molecules, the generated molecules should appear drug-like. In particular, while the generated molecules exhibit high QED (≈0.8) and favorable synthetic accessibility scores, their chemical structures raise concerns about biological and medicinal relevance. Most examples appear overly aliphatic, lacking aromaticity, heteroatoms, and functional diversity—features typically associated with bioactive compounds. As such, these molecules satisfy scalar drug-likeness metrics without reflecting realistic pharmacophoric or structural complexity. QED and SA scores, while standard, are empirical correlates of drug-likeness and can overestimate the quality of compact, hydrocarbon-rich scaffolds that are synthetically simple but biologically inert. The absence of aromatic or polar functional groups limits potential protein binding and reduces chemical interpretability. To substantiate claims of pharmacological realism, the authors should complement QED/SA evaluations with functional diversity metrics (e.g., H-bond donors/acceptors, aromatic rings), scaffold diversity analyses, and medicinal chemistry filters (e.g., Lipinski, PAINS). Incorporating bioactivity proxies such as docking or pharmacophore alignment would further strengthen the evidence for practical drug-likeness.

**Questions:**

Please see weaknesses

---

### Official Review · Reviewer_Uv6E · 2025-11-01

**Soundness:** 2
**Presentation:** 1
**Contribution:** 2
**Rating:** 2
**Confidence:** 5

**Summary:**

The paper proposes a modular, descriptor-guided patch-based GAN framework for de novo molecular generation, with a focus on antibiotic-like compounds. The architecture integrates three components: a $\beta$-VAE for learning property-aligned latent representations, a descriptor-to-latent conditioner that maps RDKit descriptors to latent embeddings, and a patch-based generator trained adversarially to synthesize molecular fragments that are later aggregated into full molecules.

The authors curate a ChEMBL subset consisting of ~4.6k antibiotic-like molecules annotated with QED, logP, and SA properties. They establish joint satisfaction reporting (QED > 0.6, SA < 5, logP $\in$ [−0.5, 5]) as a new benchmark metric. Experimental results include property distributions, stress-test robustness (50k samples), and docking/ADMET evaluations against E. coli DHFR and S. aureus DNA gyrase, demonstrating reasonable diversity and pharmacological plausibility.

Overall, the study presents an interesting and modular design for interpretable molecule generation. However, several important points concerning clarity, originality, and experimental depth need to be addressed before this work can be considered for acceptance.

The paper presents an interesting idea, but parts of the background and related work appear to be automatically written and contain factual inaccuracies (e.g., MolGAN misclassification).

**Strengths:**

***The modular structure is interesting, but text reliability is affected by some unchecked LLM-generated sections.***

**Originality:** The integration of $\beta$-VAE for latent-property conditioning within a GAN setup for molecular design is interesting. The idea of modularizing the generator into patch-level fragments offers controllable and interpretable synthesis, distinguishing it from fully atom-level GANs. The notion of joint property satisfaction (combining QED, SA, and logP thresholds) provides a valuable population-level evaluation metric for molecular generators.

**Quality:** The inclusion of 50k-sample mode collapse stress tests is commendable and demonstrates awareness of GAN stability issues. The authors report property correlations between latent dimensions and molecular descriptors (Fig. 2), showing interpretable latent–property alignment. The pipeline incorporates diverse evaluation metrics (QED, SA, logP, ADMET, FCD) and uses standard cheminformatics validation tools (SwissADME, pkCSM, RDKit).

**Experimental Breadth:** The reported Frechet ChemNet Distance (FCD = 0.1302) indicates good chemical-space alignment with ChEMBL. The model’s robustness against mode collapse (72% unique scaffolds under 50k sampling) is a notable achievement.

**Presentation and Clarity (partial):** The visualizations (e.g., violin plots, Pareto analysis, and property distributions) are informative and well-chosen. Appendices provide quantitative ablation studies, architecture diagrams, and statistical significance tests that increase transparency.

**Weaknesses:**

***Some descriptions in the background section are factually incorrect (e.g., misclassification of MolGAN as fragment-based). This indicates a lack of verification of automatically generated text. Although the authors disclosed LLM use, the unchecked factual mistakes weaken the scientific reliability of the paper.***

**Originality:** The claimed novelty overlaps substantially with prior work. The method strongly resembles [1] and employs similar components to MolGAN [4] and GraphAF [5]. The paper would benefit from clearly articulating what precise methodological advancement the $\beta$-VAE + GAN combination introduces beyond prior architectures.

The use of a graph-based discriminator is also conceptually close to prior GAN architectures in molecular generation.

*Suggestion:* Include a table or section explicitly listing what is new relative to [1], [4], and [5], e.g., “We differ by introducing X mechanism / Y conditioning / Z evaluation,” with citations to specific equations or methods.

**Clarity:** Several aspects of the methodology are insufficiently detailed or confusing:

- The rationale for choosing thresholds QED > 0.6, SA < 5, and logP $\in$ [−0.5, 5] is not explained; a literature justification (e.g., based on ADMET standards) should be cited.
- In Figure 1, the data flow between $\beta$-VAE, conditioner, and generator is semantically unclear. The diagram should be restructured to distinguish training from inference stages.
- Table 1 lists baseline models, but the connection to fragment-based approaches is ambiguous. For example, methods [2] and [3] are already fragment-based, contrary to what is suggested.
- The notation in Section~3.2 could be improved. For instance, the dimensionality term $d$ in $X \in \mathbb{R}^{n \times d}$ is undefined, and the variable selection procedure ("we remove invalid features...'') lacks justification.
- The "best GAN-trained model'' section introduces a custom reward function:

$ \mathrm{Reward} = 2\ \times \mathrm{QED} - 0.6\ \times \max\\bigl(SA-3,0\bigr)^{4} - 0.4\ \times \max\\bigl(\lvert \log P - 2.5\rvert - 1,0\bigr)^{2}. $

However, the choice of coefficients \(2, 0.6, 0.4\) appears ad hoc and is not motivated. Please provide empirical or theoretical reasoning (e.g., ablations or a principled weighting scheme) to justify these values.

*Suggestion:* A small ablation study over these coefficients (or a sensitivity analysis) could strengthen the credibility of the model-selection procedure.

**Evaluation:**
- **Dataset handling:** The ChEMBL curation process is not fully transparent—what was the train/validation/test split? Were docking molecules also used for training? Without such details, overfitting or data leakage cannot be ruled out.
- **Performance comparison:** Table 3’s baseline results (InstGAN, L-MolGAN, etc.) are quoted from original papers, not re-evaluated on the same dataset. Hence, comparisons may not be fair.
- **Statistical rigor:** Results should include p-values or confidence intervals, especially in Fig. 4’s property distributions and docking metrics.
- **Biological validation:** Docking was performed on only 281 ligands (DHFR) and 9 outperforming cases (DNA gyrase). It is unclear why these specific subsets were chosen, and whether they were representative.
- **3D structure realism:** The paper repeatedly uses terms like “structural realism”, yet all operations are in 2D graph space. Without 3D conformation or geometric checks, this claim seems overstated.

*Suggestion:* Replace “structural realism” with “topological realism” unless 3D geometries are directly modeled, or add a 3D-based evaluation (e.g., RMSD or alignment metrics).

**Conceptual and Referencing Issues:** Some background statements may misrepresent prior works. For example, MolGAN [4] and Hoogeboom et al. [5] are not fragment-based methods, but the text implies otherwise.

References to SwissADME and pkCSM heuristics should be properly cited and linked in the main text, not only mentioned in the appendix.
Certain claims (e.g., “this loop enhances structural realism” or “we implemented an ADMET scoring script”) should include appropriate citations or methodological details.

**Adjustment Suggestions:**

Figure 1 is difficult to interpret due to dense labeling. Consider splitting it into two: (a) architecture overview and (b) training workflow.

Typographical errors: “antibiotov-like,” “combining,” “mnodel,” and similar typos appear frequently. A thorough proofreading is recommended.

**References**

[1] https://link.springer.com/article/10.1186/s13321-025-00965-x

[2] https://www.biorxiv.org/content/10.1101/2022.11.21.517375v1

[3] https://arxiv.org/abs/2401.05370

[4] https://arxiv.org/abs/1805.11973

[5] https://www.nature.com/articles/s42256-025-01082-y

**Questions:**

- **Could the authors clarify which parts of the text were generated or polished by LLMs, and what verification steps were applied?**
- **Will the authors correct the inaccurate claims (e.g., MolGAN being fragment-based) in the final version?**

- Could you show quantitative results comparing your model directly with GraphAF or MolGAN on the same dataset?
- What motivates the specific coefficients in your custom reward functions? Were they tuned empirically?
- In Figure 2, why do correlations drop when latent dimensionality increases?
- How did you prevent data leakage when curating from ChEMBL?
- Could you report p-values for the distributions shown in violin plots (Fig. 4)?
- How do you handle potential decoys among high-affinity docking results?
- Why were only 281 ligands docked for DHFR and only 9 surpassing ciprofloxacin reported?
- Did you attempt benchmarking with other property-conditioned GANs from [6]?
- How do you justify calling your generator “fragment-based” if no explicit fragment vocabulary or decomposition mechanism is used?
- Could you share quantitative evidence that your generated molecules maintain “controllable latent graph embeddings”?

**References:**

[1] https://www.nature.com/articles/s42256-025-01082-y/tables/1

**Details Of Ethics Concerns:**

The authors disclose limited use of LLMs for text polishing, but parts of the introduction and background sections appear to include factual inaccuracies likely caused by unchecked automated rewriting.

For instance, MolGAN[1] is incorrectly described as a fragment-based model, which it is not. While this does not constitute plagiarism or data misuse, it raises minor concerns regarding responsible research practice — specifically, the need for authors to fact-check and verify statements produced by language models before submission.

I am flagging this as a mild research integrity concern rather than a serious ethical violation.

**References:**

**[1]** CaoN, Kipf T., MolGAN: An implicit generative model for small molecular graphs. 2018, arXiv:1805.11973

---

### Meta-Review · Area_Chair_33G1 · 2026-01-07

**Summary:**

The reviewers raised several consistent concerns across all four reviews that informed my rejection recommendation:
1. All reviewers noted the extremely low 2.85% joint multi-property satisfaction (MPS) rate as problematic, with only 464 molecules meeting the stated constraints from over 16,000 generated. Reviewers n8pz and SaTA specifically flagged that the generated molecules in Fig. 11 lack drug-like features—particularly aromatic rings and heteroatoms—despite high QED scores, questioning whether they genuinely resemble viable antibiotic candidates. The reliance entirely on in-silico validation without experimental verification limits confidence in practical applicability.
2. Multiple reviewers (n8pz, SaTA, tpxG) identified that Table 3 lacks MPS scores for baseline models, making fair comparison impossible. While the authors introduce MPS as their key metric, they don't demonstrate superiority over alternatives under this metric. The reviewers were unconvinced by the authors' argument that retraining baselines would be unfair—establishing a new benchmark requires showing it works better.
3. Reviewers Uv6E and tpxG noted substantial overlap with prior work on descriptor conditioning (Mol-CycleGAN, InstGAN), fragment-based generation (DeepMGM, JT-VAE), and VAE-GAN combinations. The distinction between "patches" and existing fragment vocabularies was unclear. Reviewer tpxG stated the paper "mainly combines existing components" without fundamental algorithmic innovation.
4. Reviewer Uv6E identified factual errors including MolGAN misclassification, likely from unchecked LLM-generated text. Multiple methodological issues were raised: lack of proper train/test splits for GAN evaluation, unjustified threshold choices, unclear reward function coefficients, and conflation of 2D topology with 3D "structural realism." The small dataset (4,607 molecules) without broader generalization experiments raised overfitting concerns.
5. Several critical issues—particularly the aromaticity problem and low MPS—were only addressed after submission through aggregator improvements, suggesting the work was not mature enough for publication at submission time.

**Reviewer Concerns:**

The authors successfully addressed several technical clarifications. The MolGAN misclassification (Reviewer Uv6E Q2) was acknowledged as an editorial oversight from compressing sections, not a scientific error, and corrected. The reward coefficient justification (Uv6E Q4) was explained with variance-scaling rationale and sensitivity analysis showing <2% ranking change. The correlation versus latent dimensionality question (Uv6E Q5) was clarified as variance redistribution rather than loss, with cumulative correlations remaining stable. Statistical rigor concerns (Uv6E Q7) were addressed by adding p-values in appendices. The docking subset selection (Uv6E Q9) was explained as focusing on top-ranked compounds by composite scores. The "fragment-based" terminology confusion (Uv6E Q11) was resolved by adopting "patch-based" throughout. The frozen discriminator choice (SaTA W3) was justified based on low-entropy generator characteristics causing instability during co-training. The SA computation runtime constraint (SaTA Q1) was explained as stemming from full sanitization requirements, not the SA function itself. The train/test split concern (Uv6E Q6) was appropriately defended as following standard unsupervised GAN evaluation practices in molecular generation.

The most critical concern—the 2.85% MPS rate (Reviewers n8pz W3, SaTA W1, tpxG W5)—was attributed to aggregator limitations fixed post-submission. While the authors claim improvements, this represents a fundamental functional issue that should have been resolved before submission. The reviewers legitimately questioned whether the work was ready for publication. The missing baseline MPS comparisons (n8pz W2, SaTA W2, tpxG W4) remain problematic. The authors explain that baselines don't release necessary data and retraining would be unfair, but this essentially means they introduce a metric without demonstrating superiority over alternatives—undermining the paper's core contribution. The aromaticity issue (n8pz W3, SaTA Q3) revealed molecules lacking basic drug-like features, again fixed only post-submission. The novelty concerns (Uv6E originality weakness, tpxG W6) were addressed with a promised comparison table, but the fundamental question of whether combining existing components constitutes sufficient contribution remains. The small dataset and lack of generalization (tpxG W1) was defended as domain-appropriate, but internal QM9 tests mentioned only post-submission suggest the work may be limited to the specific training set. The joint threshold justification (n8pz W1) acknowledged overlap with Pandey et al. 2025 but didn't fully resolve whether the specific constraints represent genuine novelty. The biological validation limitation (SaTA W4) was acknowledged as in-silico only, but for a paper claiming antibiotic discovery applications, this remains a significant gap between claims and evidence.

In summary, technical clarifications were handled well, but fundamental performance issues, missing comparative benchmarking, and post-submission fixes to core functionality represent unresolved concerns that justify rejection.

**Reviewer Scores:**

Reviewer Uv6E (initial score: 2 - reject): Likely would remain at 2. This reviewer was the most critical, flagging factual inaccuracies and LLM-generated text issues.

Reviewer n8pz (initial score: 2 - reject): Would likely remain at 2. This reviewer's three main concerns were: (1) threshold claims overlapping with Pandey et al., (2) missing baseline MPS values, and (3) non-drug-like molecules lacking aromaticity. The first was acknowledged but not fully resolved. The second remains fundamentally unaddressed—the authors explain why they can't provide it but this doesn't solve the fairness problem. The third was attributed to post-submission aggregator fixes, which might not satisfy a reviewer who expects submission-ready work. The core benchmarking concern remains outstanding.

Reviewer SaTA (initial score: 4 - marginally below acceptance): Could potentially increase to 6. This was the most positive reviewer, already scoring near the acceptance threshold. The authors provided reasonable responses to most concerns.

Reviewer tpxG (initial score: 4 - marginally below acceptance): Would likely remain at 4. This reviewer raised concerns about dataset size, missing baseline comparisons, unclear novelty, and low MPS. The authors provided reasonable defenses for dataset choice and β-VAE's role, but the fundamental novelty question and missing baseline benchmarking remain unresolved.

---

### Decision · Program_Chairs · 2026-01-26

Reject